# Starting to have sexual intercourse is associated with increases in cervicovaginal immune mediators in young women: a prospective study and meta-analysis

Sean M Hughes[1], Claire N Levy[1], Fernanda L Calienes[2], Katie A Martinez[1], Stacy Selke[3], Kenneth Tapia[4], Bhavna H Chohan[4,5], Lynda Oluoch[6], Catherine Kiptinness[6], Anna Wald[2,3,7,8], Mimi Ghosh[9], Liselotte Hardy[10], Kenneth Ngure[4,11], Nelly R Mugo[4,6], Florian Hladik[1,2,7]*†, Alison C Roxby[2,4,7,8]*†

[1]Department of Obstetrics and Gynecology, University of Washington, Seattle, United States; [2]Vaccine and Infectious Disease Division, Fred Hutchinson Cancer Research Center, Seattle, United States; [3]Department of Laboratory Medicine & Pathology, University of Washington, Seattle, United States; [4]Department of Global Health, University of Washington, Seattle, United States; [5]Centre for Virus Research, Kenya Medical Research Institute, Nairobi, Kenya; [6]Centre for Clinical Research, Kenya Medical Research Institute, Nairobi, Kenya; [7]Department of Medicine, University of Washington, Seattle, United States; [8]Department of Epidemiology, University of Washington, Seattle, United States; [9]Department of Epidemiology, Milken Institute School of Public Health, The George Washington University, Washington, United States; [10]Department of Clinical Sciences, Unit of Tropical Bacteriology, Institute of Tropical Medicine, Antwerp, Belgium; [11]Department of Community Health, Jomo Kenyatta University of Agriculture and Technology, Nairobi, Kenya

*For correspondence: florian@uw.edu (FH); aroxby@uw.edu (ACR)

†These authors contributed equally to this work

## Abstract

**Background:** Adolescent girls and young women (AGYW) are at high risk of sexually transmitted infections (STIs). It is unknown whether beginning to have sexual intercourse results in changes to immune mediators in the cervicovaginal tract that contribute to this risk.

**Methods:** We collected cervicovaginal lavages from Kenyan AGYW in the months before and after first penile-vaginal sexual intercourse and measured the concentrations of 20 immune mediators. We compared concentrations pre- and post-first sex using mixed effect models. We additionally performed a systematic review to identify similar studies and combined them with our results by meta-analysis of individual participant data.

**Results:** We included 180 samples from 95 AGYW, with 44% providing only pre-first sex samples, 35% matched pre and post, and 21% only post. We consistently detected 19/20 immune mediators, all of which increased post-first sex (p<0.05 for 13/19; Holm-Bonferroni-adjusted p<0.05 for IL-1β, IL-2, and CXCL8). Effects remained similar after excluding samples with STIs and high Nugent scores. Concentrations increased cumulatively over time after date of first sex, with an estimated doubling time of about 5 months.

Our systematic review identified two eligible studies, one of 93 Belgian participants, and the other of 18 American participants. Nine immune mediators were measured in at least two-thirds of studies. Meta-analysis confirmed higher levels post-first sex for 8/9 immune mediators ($p < 0.05$ for six mediators, most prominently IL-1α, IL-1β, and CXCL8).

**Conclusions:** Cervicovaginal immune mediator concentrations were higher in women who reported that they started sexual activity. Results were consistent across three studies conducted on three different continents.

**Funding:** This research was funded by R01 HD091996-01 (ACR), by P01 AI 030731-25 (Project 1) (AW), R01 AI116292 (FH), R03 AI154366 (FH) and by the Center for AIDS Research (CFAR) of the University of Washington/Fred Hutchinson Cancer Research Center AI027757.

## Editor's evaluation

This study finds that the levels of many immune markers are higher in vaginal samples in women taken after initiation of vaginal sex than before initiation of vaginal sex. This result may indicate that initiation of vaginal sex potentially influences vaginal immune responses in adolescents and young adults. This study will be of the highest interest to those interested in how immune markers can change within individuals over time.

## Introduction

Adolescent girls and young women (AGYW) aged 15–24 are at high risk for sexually transmitted infections (STIs) and are disproportionately affected by human immunodeficiency virus (HIV), accounting for as many as 80% of new HIV infections in some countries (*UNAIDS, 2019*). In Kenya, in particular, a quarter of new HIV infections occur in AGYW (*National AIDS Control Council, 2018*). Because of the disproportionate impact of HIV on AGYW, understanding the behavioral and physiological components of heightened STI risk in this population presents an opportunity to reduce HIV infections.

The period following first penile-vaginal sexual intercourse ("first sex") marks the start of vulnerability to STIs and is associated with higher acquisition of bacterial vaginosis (BV) and STIs than later in life. The reasons for this increased susceptibility remain unclear, because (1) studies of mucosal immunity in AGYW are challenging, and (2) it can be difficult to distinguish immune changes that are a consequence of STI acquisition from changes that occur independently of STIs. Understanding changes in cervicovaginal tract (CVT) immune mediators following first sex may help identify interventions to decrease the risk of STI and HIV acquisition in AGYW.

In this study, we measured immune mediators in cervicovaginal lavage (CVL) specimens collected from a unique longitudinal cohort of Kenyan AGYW (*Casmir et al., 2021*; *Yuh et al., 2020*), an especially vulnerable population. By comparing specimens collected in the months before and after first sex, our goal was to measure changes in CVT immune mediators following start of sexual intercourse. Extensive clinical information was available about participants, including acquisition of STIs.

To generalize our findings to broader populations, we also conducted a systematic review of published literature to identify other studies of cervicovaginal immune mediators in AGYW before and after first sexual intercourse. We sought individual participant data from study authors and performed meta-analyses of individual immune mediators, assessing changes in immune mediator concentration before and after first sex.

## Materials and methods

**Key resources table**

| Reagent type (species) or resource | Designation | Source or reference | Identifiers | Additional information |
|---|---|---|---|---|
| Biological sample (*Homo sapiens*) | Cervicovaginal lavage samples | This paper | | The dataset is included in *Source code 1* |
| Commercial assay or kit | Quantifiler Duo DNA Quantification Kit | Thermo Fisher | Catalog# 4387746 | |

*Continued on next page*

*Continued*

| Reagent type (species) or resource | Designation | Source or reference | Identifiers | Additional information |
|---|---|---|---|---|
| Commercial assay or kit | Pierce BCA Protein Assay Kit | Thermo Fisher | Catalog# 23225 | |
| Commercial assay or kit | MIG R-Plex assay | Meso Scale Discovery | Catalog# F210I | |
| Commercial assay or kit | RANTES R-Plex assay | Meso Scale Discovery | Catalog# F21ZN | |
| Commercial assay or kit | IL-1RA U-Plex assay | Meso Scale Discovery | Catalog# K151XPK | |
| Commercial assay or kit | IFN-γ U-Plex assay | Meso Scale Discovery | Catalog# K151TTK | |
| Commercial assay or kit | IL-10 U-Plex assay | Meso Scale Discovery | Catalog# K151TZK | |
| Commercial assay or kit | IL-12p70 U-Plex assay | Meso Scale Discovery | Catalog# K151UAK | |
| Commercial assay or kit | IL-1β U-Plex assay | Meso Scale Discovery | Catalog# K151TUK | |
| Commercial assay or kit | IL-2 U-Plex assay | Meso Scale Discovery | Catalog# K151TVK | |
| Commercial assay or kit | IL-6 U-Plex assay | Meso Scale Discovery | Catalog# K151TXK | |
| Commercial assay or kit | IL-8 U-Plex assay | Meso Scale Discovery | Catalog# K151TYK | |
| Commercial assay or kit | TNF-α U-Plex assay | Meso Scale Discovery | Catalog# K151UCK | |
| Commercial assay or kit | IFN-a2a U-Plex assay | Meso Scale Discovery | Catalog# K151VHK | |
| Commercial assay or kit | IL-17a U-Plex assay | Meso Scale Discovery | Catalog# K151UTK | |
| Commercial assay or kit | IL-18 U-Plex assay | Meso Scale Discovery | Catalog# K151VJK | |
| Commercial assay or kit | IL-1α U-Plex assay | Meso Scale Discovery | Catalog# K151UNK | |
| Commercial assay or kit | IL-7 U-Plex assay | Meso Scale Discovery | Catalog# K151UPK | |
| Commercial assay or kit | IP-10 U-Plex assay | Meso Scale Discovery | Catalog# K151UFK | |
| Commercial assay or kit | MIP-1a U-Plex assay | Meso Scale Discovery | Catalog# K151UJK | |
| Commercial assay or kit | MIP-1β U-Plex assay | Meso Scale Discovery | Catalog# K151UKK | |
| Commercial assay or kit | MIP-3α U-Plex assay | Meso Scale Discovery | Catalog# K151UZK | |
| Commercial assay or kit | Human Kallikrein 3/PSA DuoSet ELISA | R&D Systems | Catalog# DY1344 | |
| Software, algorithm | R Project for Statistical Computer | R Core Team | RRID:SCR_001905 | |
| Software, algorithm | R package: lmerTest | Comprehensive R Archive Network | RRID:SCR_015656 | |
| Software, algorithm | R package: meta | Comprehensive R Archive Network | RRID:SCR_019055 | |
| Software, algorithm | R package: tidyverse | Comprehensive R Archive Network | RRID:SCR_019186 | |
| Other | Ghosh 2018 dataset | *Ghosh et al., 2018* | | The dataset is included in *Source code 1* |
| Other | Jespers 2016 dataset | *Jespers et al., 2016* | | The dataset is included in *Source code 1* |

## Clinical cohort and study procedures

This study used specimens collected from the Kenya Girls Study, a previously described longitudinal cohort study of AGYW (*Casmir et al., 2021*; *Yuh et al., 2020*). Briefly, AGYW aged 16–20 were recruited in Thika, Kenya. Enrolled participants returned quarterly, where they were interviewed about their sexual behavior and provided vaginal swabs, CVLs, and blood. For the sub-study described in this paper, we selected a group of 195 samples from the larger Kenya girls study, including samples from participants who provided only pre-first sex samples, participants who provided only post-first sex samples, and participants who provided both pre- and post-first sex samples. The sample size was chosen based on selection of all available samples from all participants who reported sexual intercourse and a matching number of participants who did not.

Human subjects approval was obtained from the Kenya Medical Research Institute Scientific Ethics Review Unit (protocol 2760) and the University of Washington Institutional Review Board (number 00000946). Participants under age 18 provided written informed assent, and written informed consent was obtained from a parent/guardian. Participants assented privately from parents/guardians, after asking questions and deciding whether they wanted to participate free of parental influence. Participants age 18 or older provided written informed consent.

CVLs were collected using flexible tubing in the vagina to avoid a speculum examination. 5 cc of sterile saline was instilled into the vagina via tubing by the study clinician, left for 15 s, then aspirated into the syringe. Lavage fluid was spun for 10 min at 800×g; supernatant was removed, re-spun for 10 min, and stored in 2 mL aliquots at –80°C.

Participants were tested for STIs and BV. Vaginal swabs were tested for NG, CT, and TV using the Gen-Probe Aptima test (Hologic, Marlborough, MA) and for HSV-1 and HSV-2 by in-house PCR. Nugent scoring was performed on smears from vaginal swabs for BV; scores ≥7 were considered BV, scores 4–6 were considered intermediate, and scores ≤3 were considered no dysbiosis (*Nugent et al., 1991*). Blood was tested for HIV using the Vironostika HIV Uni-Form II Ag-Ab (Biomerieux, Marcy-l'Etoile, France) and for HSV-1 and HSV-2 antibodies using in-house Western blot.

STI tests were performed annually in the parent study. Therefore, STI testing had not been performed at some timepoints selected for this sub-study. For samples with no concurrent STI result, we determined STI status as follows: If the annual tests both before and after the untested visit were negative, we inferred a negative result for that visit. If either annual test was positive, STI testing was performed for that timepoint. Negative CT and NG tests were inferred for 28 samples as well as negative TV tests for 45 samples.

Participants were tested for pregnancy via rapid urinary pregnancy test if they reported missed menses. Serum samples were tested for progesterone by automated immunoassay (Cobas E411, Roche, Basel, Switzerland). Participants were defined to be in the follicular phase if serum progesterone was <3 ng/mL, in the luteal phase if serum progesterone was ≥3 ng/mL, and 'other' if more than 35 days since the start of their last menstrual period had passed (indicating irregular menstruation, use of hormonal contraception, or pregnancy). Participants were assumed not to be using contraception until they reported sexual intercourse.

PSA was measured in CVL using the Human Kallikrein 3/PSA DuoSet ELISA (R&D Systems, Minneapolis, MN). Samples were considered positive if the PSA concentration was ≥10 ng/mL. Total protein concentrations were measured using the Pierce BCA Protein Assay Kit (Thermo Fisher Scientific, Waltham, MA). PSA and total protein concentrations were calculated using four-parameter logistic curves fit to the standard curves. Vaginal swabs were tested for Y-chromosome DNA by Quantifiler Duo DNA Quantification Kit (Life Technologies).

The immune mediator data reported in this article was previously included in a meta-analysis of changes in immune mediator concentrations during the menstrual cycle (*Hughes et al., 2022*), where it is referred to as "Hughes-unpublished".

## Immune mediator quantification

We selected 20 immune mediators to measure in CVLs. These immune mediators were selected based on prior human and/or non-human primate studies as being consistently up- or down-regulated prior to HIV or STI transmission events, and therefore playing a key role in genital tract immunity to STIs. Concentrations of immune mediators were measured in CVL using Meso Scale Discovery (MSD) R-Plex/U-Plex kits according to the manufacturer's instructions and read on the MESO QuickPlex SQ 120. Pre- and post-first sex samples were present on every plate, and the scientists were blinded to sample identity. CVLs were diluted 100-fold for measurement of IL-1RA and 10-fold for CXCL9 and CCL5. Concentrations were determined using four parameter logistic fits in MSD Discovery Workbench software.

## Definition of first sexual intercourse

Samples were categorized as pre- or post-first sex based on participant report of ever engaging in penile-vaginal penetrative sex. In addition, visits were categorized as post-first sex if at that visit or a previous visit, the participant was pregnant, Y-chromosome DNA was detected in a vaginal swab sample, PSA was detected in a CVL, or a vaginal swab tested positive for NG, CT, or TV.

## Statistical analysis

Data analysis was conducted using R (*R. Core Team, 2018*) version 4.0.0 with the packages plater (*M Hughes, 2016*) and tidyverse (*Wickham et al., 2019*) in addition to the statistical packages described below.

For MSD, concentrations of each analyte were averaged across replicate wells and log2-transformed. Replicates with concentrations below the reported lower limits of detection were assigned the value of half the lower limit of detection. Samples were defined as detectable if the concentration was above the lower limit of detection for at least one of the two replicate wells. Some samples were excluded due to rare STIs or missing data (fully described in *Figure 1*), these exclusion criteria were not pre-established.

We performed primary and secondary analyses. For the primary analysis, we used univariate models to estimate the difference between cytokine concentrations in pre- and post-first sex samples (assessed by effect size and p-value). Pre- and post-first sex were modeled as a binary fixed effect and participant as a random effect to account for multiple samples per participant using mixed-effect models with the R packages lme4 (*Bates et al., 2015*) and lmerTest (*Kuznetsova et al., 2017*). Log2-transformed immune mediator concentrations were used as the outcome for immune mediators if more than 50% of samples were above the limit of detection. For the remaining immune mediators, logistic models were used with the outcome being above/below the lower limit of detection. Logistic models were used for immune mediators with low levels of detection because the distributions become increasingly non-normal as more samples fall below the lower limit of detection. We adjusted for multiple comparisons by the Holm-Bonferroni method (*Holm, 1979*).

We performed several secondary analyses. We tested whether pregnancy, STIs, BV, and contraception explained our results by repeating our primary, univariate analysis on the subset of samples negative for all of these variables. In addition, we repeated our univariate models using immune mediator concentrations normalized to total protein. Furthermore, we performed univariate models on the subset of samples from participants who provided both pre- and post-first sex samples. Finally, we fit multivariate models with additional fixed effects: age, education at enrollment, rural/urban residence at enrollment, and roof material at enrollment (an indicator of poverty). In all cases, we assessed whether these factors had a major influence on results by comparing the effect sizes from the primary analysis to the effect sizes from the secondary analysis.

We additionally wished to assess whether immune mediator concentrations changed over time with respect to the date of first sex. To assess this question, we fit mixed-effect models with linear splines, using days since first sex as a fixed effect and participant as a random effect, with the outcome being the concentration of immune mediator (log2 pg/mL). For the splines, we used a knot at day 0, which was the day of first sex.

All code and raw data are available in *Source code 1* and on GitHub at https://github.com/seaaan/ThikaPrePostSexManuscript/ (copy archived at swh:1:rev:ef3bd4bee186be79944ebd821fd5e1b-b2e89a26b; *Hughes, 2022*). All results are available in *Supplementary file 1*.

## Systematic review and meta-analysis

We designed our systematic review and meta-analysis in compliance with the preferred reporting items for a systematic review and meta-analysis of individual participant data (PRISMA-IPD) guidelines (*Stewart et al., 2015*). The PRISMA-IPD checklist is included as *Reporting standard 1*.

### Study eligibility criteria

Studies were eligible if they reported original immunoassay data on immune mediator concentrations in CVT fluid samples from AGYW before and after first vaginal sexual intercourse. Immunoassay data included any antibody-based methods (such as ELISA, bead-based assays, and electrochemiluminescence assays). CVT fluid samples could be collected by CVL, menstrual cup, or swab. Each study was required to include samples from before and after first penile-vaginal sexual intercourse, but we did not require that there be pre- and post-first sex samples from each participant. Samples were required to be categorized as pre- or post-first sex based on participant report of ever engaging in penile-vaginal penetrative sex. In addition, visits were categorized as post-first sex based on pregnancy, detection of sperm antigen or DNA, or a positive STI test.

### Study identification and inclusion

We searched PubMed on August 28, 2021 for articles published in English using the search strategy in *Supplementary file 2*. Abstracts from search results were reviewed independently by two reviewers (SMH and CNL) using abstrackr (*Wallace et al., 2012*). If either reviewer judged the study to be potentially eligible based on the abstract, then full-text articles were obtained and reviewed by both reviewers. Study inclusion was determined by consensus between the two reviewers.

### Data collection and verification

We sought individual participant data via email with the authors of eligible studies. For each sample, we collected the participant identifier, the pre/post-first sex status of the sample, and the immune mediator concentrations in pg/mL. For each study, we collected the type of immunoassay used, the type of sample collected, and the country of sample origin. Individual participant data were verified by replicating analyses published in the prior manuscripts. Pre/post-first sex status was standardized across studies: in particular, one study included samples from participants reporting genital touching without penile-vaginal sexual intercourse. We categorized these samples as pre-first sex in accordance with our definition above. No personally identifiable information was obtained from the studies included in the meta-analysis. Participants in both studies provided informed consent and consent to publish as described in the original manuscripts (*Jespers et al., 2016*; *Ghosh et al., 2018*).

### Data analysis

The outcome of interest was the difference in immune mediator concentrations between pre- and post-first sex samples. We performed meta-analyses on all immune mediators present in at least two included studies. We used a two-stage approach for meta-analysis. First, we reanalyzed the individual participant data from the eligible studies. Studies with multiple samples per participant were analyzed using mixed-effect models as above; studies with single samples per participant were analyzed using simple linear models. For all studies, immune mediator concentrations (pg/mL) were log2-transformed. In the second stage, we performed random-effects meta-analysis using the inverse-variance method in the R package meta (*Balduzzi et al., 2019*). We quantified heterogeneity using $I^2$. We did not attempt to explore variation in effects by study-level characteristics due to the low number of studies available.

### Risk of bias

We assessed risk of bias within each study using a modified Newcastle-Ottawa Quality Assessment scale (*Supplementary file 3*), ranging from 0 (high risk of bias) to 7 (low risk of bias).

## Results

### Characteristics of cohort

We selected 195 CVL samples for this study. We excluded 15 samples due to unavailable covariate information (such as total protein or Prostate-specific antigen [PSA]), extensive blood contamination, and presence of rare infections (n=1–2 samples per infection: *Trichomonas vaginalis* (TV), *Neisseria gonorrhoeae* (NG), genital HSV-1 or HSV-2 DNA; *Figure 1A*). The final sample set included 180 samples from 95 participants.

Of these, 111 samples from 75 participants were classified as pre-first sex, and 57 samples from 45 participants were obtained after reported first sex. In addition, 12 samples from 8 participants were classified as post-sex due to presence of Y-chromosome DNA, PSA, or because of pregnancy or a positive *Chlamydia trachomatis* (CT) test at that visit or a prior visit. Thus, there were a total of 69 post-sex specimens from 53 participants. Pre-first sex samples were collected a median of 344 days before first sex (interquartile range [IQR] 72–687), while post-first sex samples were collected a median of 66 days (27-105) after. We captured matched pre- and post-first sex specimens from 33 participants (66 specimens); the remaining participants provided either only pre-first sex specimens (42 participants; 78 specimens) or only post-first sex specimens (20 participants; 36 specimens; *Figure 1B*).

*Table 1* summarizes demographic and clinical variables of the samples from the final sample set. The median age of participants in the final sample set was 19.1 years (IQR 18.1–19.6). All participants

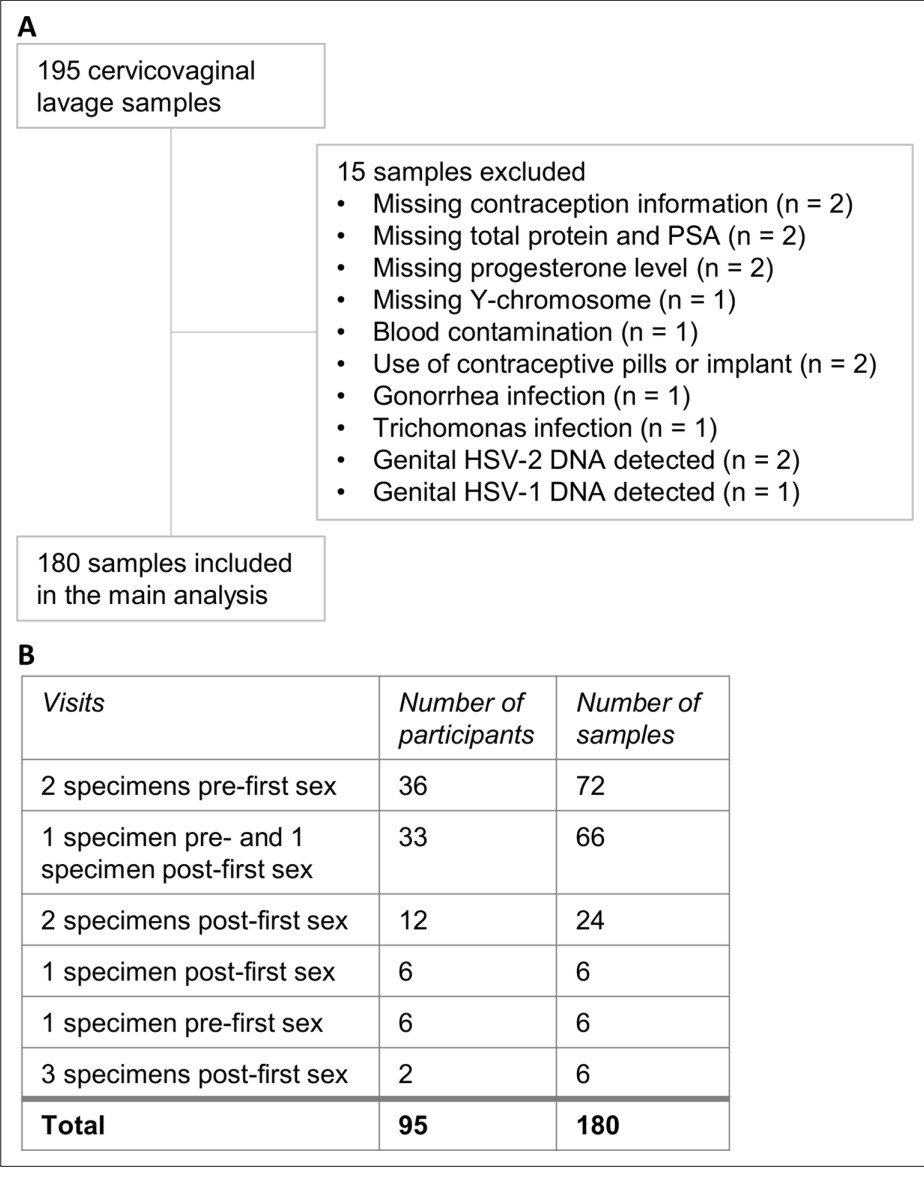

**Figure 1.** Cervicovaginal lavage sample selection. (**A**) Cervicovaginal lavage sample selection, collected from adolescent girls and young women in a longitudinal cohort study. (**B**) Timepoints for collection of cervicovaginal lavage specimens pre- and post-first sexual intercourse.

were Black Africans assigned female at birth. Post-first sex samples came from participants who were, on average, older and more likely to have BV (*Table 1*). By definition, only the post-first sex group included samples from pregnant participants, participants using contraception, and samples where Y-chromosome, PSA, CT, or HSV-2 was detected.

## Immune mediator measurements pre- and post-first sex

Concentrations of 20 immune mediators were measured (*Figure 2*). IL-1RA was detected at the highest concentrations. Sixteen immune mediators were detectable in most specimens. Four were detected in fewer than half of the samples: IFN-γ (27%), IL-10 (23%), IL-12p70 (20%), and IFN-α2A (7%); due to lack of detection, IFN-α2A was excluded from further analysis.

As shown in *Figure 2*, immune mediator concentrations were higher after first sex. We quantified the differences between pre- and post-first sex samples (*Figure 2*). In all cases, the concentration (*Figure 3A*, *Table 2*) or proportion detectable (*Figure 3B*, *Table 3*) was higher. The differences were

**Table 1.** Comparison of social, demographic, and biological characteristics of adolescent girls and young women at timepoints of cervicovaginal lavage specimens selected for the study.

Each participant can appear in this table more than once. Pregnant indicates pregnant at the time of sample collection. All participants listed here were negative for HSV-1 and HSV-2 genital DNA (indicative of active infection). Mixed effect regression models included a random intercept for participant.

| | Pre-first sex timepoints N=111 specimens* | Post-first sex timepoints N=69 specimens* | p-Value |
|---|---|---|---|
| Age (years) | 18.80 (17.80, 19.45) | 19.30 (18.60, 19.80) | <0.001† |
| Education at enrollment‡ | | | 0.934§ |
| Completed high school | 18 (16%) | 12 (17%) | |
| In high school | 93 (84%) | 57 (83%) | |
| Income at enrollment‡ | | | 0.876§ |
| No regular income | 57 (51%) | 43 (62%) | |
| Regular income | 54 (49%) | 26 (38%) | |
| Residence at enrollment‡ | | | 0.575§ |
| Rural | 72 (65%) | 28 (41%) | |
| Urban | 39 (35%) | 41 (59%) | |
| Roof material at enrollment‡ | | | 0.814§ |
| Not tile or metal | 10 (9%) | 1 (1.4%) | |
| Tile or metal | 101 (91%) | 68 (99%) | |
| Menstrual phase | | | |
| Luteal | 39 (35%) | 26 (38%) | 0.73§ |
| Follicular | 64 (58%) | 35 (51%) | 0.36§ |
| Other | 8 (7.2%) | 8 (12%) | 0.21§ |
| Pregnant | | | –¶ |
| Negative | 111 (100%) | 64 (93%) | |
| Positive | 0 (0%) | 5 (7.2%) | |
| Contraception | | | –¶ |
| None | 111 (100%) | 54 (78%) | |
| Emergency pills | 0 (0%) | 4 (5.8%) | |
| Condoms | 0 (0%) | 11 (16%) | |
| Y-chromosome | | | –¶ |
| Negative | 111 (100%) | 55 (80%) | |
| Positive | 0 (0%) | 14 (20%) | |
| Prostate-specific antigen | | | –¶ |
| Negative | 111 (100%) | 67 (97%) | |
| Positive | 0 (0%) | 2 (2.9%) | |
| Bacterial vaginosis | | | |
| Negative (Nugent score 0–3) | 102 (92%) | 54 (78%) | 0.08§ |
| Intermediate (4–6) | 6 (5.4%) | 6 (8.7%) | 0.82§ |
| Positive (7–10) | 3 (2.7%) | 9 (13%) | 0.02§ |

*Table 1 continued on next page*

*Table 1 continued*

| | Pre-first sex timepoints N=111 specimens[*] | Post-first sex timepoints N=69 specimens[*] | p-Value |
|---|---|---|---|
| Chlamydia | | | -[¶] |
| Negative | 111 (100%) | 59 (86%) | |
| Positive | 0 (0%) | 10 (14%) | |
| HSV-1 | | | >0.9[§] |
| Seronegative | 4 (3.6%) | 2 (2.9%) | |
| Seropositive | 107 (96%) | 67 (97%) | |
| HSV-2 | | | -[¶] |
| Seronegative | 111 (100%) | 66 (96%) | |
| Seropositive | 0 (0%) | 3 (4.3%) | |

[*]Statistics presented: median (IQR) or n (%).
[†]Linear mixed effects regression.
[‡]Time-invariant variables, with data only available from the enrollment visit.
[§]Logistic mixed effects regression with binary combinations, comparing each category to all other samples (e.g. luteal compared to pooled follicular and other).
[¶]Not tested for difference between groups because this variable can only be positive at post-first sex time points by definition.

significant at $p<0.05$ for most (13/19) immune mediators and remained significant for IL-1β, IL-2, and CXCL8 after adjustment for multiple comparisons (adjustment for 19 immune mediators).

## Immune mediator concentrations over time

We next assessed whether immune mediator concentrations increased all at once at first sex or cumulatively over time. We therefore evaluated immune mediator concentrations relative to date of first sex. Dates were available for 80 pre-first sex samples from 59 participants and 60 post-first sex samples from 49 participants.

As shown in *Figure 4* and *Table 4*, immune mediator concentrations were generally stable for 3 years prior to first sex and then increased sharply in the year following first sex. The change in slope following first sex was statistically significant at $p<0.05$ for 17/19 immune mediators. This pattern is consistent with cumulative increases in immune mediator concentrations following first sex.

## Assessment of other contributing factors

We next sought to determine whether the differences between pre- and post-first sex could be explained by covariates that differed between the groups or that are known to affect immune mediator concentrations (*Table 1*). In particular, pregnancy, CT infection, HSV-2 seropositivity, use of contraception, and elevated Nugent score are all known to affect cervicovaginal immune mediator concentrations. Because of this, we wished to know whether one or more of these factors was responsible for the differences we observed between pre- and post-first sex samples. To address this possibility, we repeated our analysis on a restricted sample set, excluding all samples where at the visit the participant was pregnant, positive for CT or HSV-2, reported using contraception, or had a Nugent score of 4 or greater. The analysis of these 137 samples (102 pre- and 35 post-first sex) is shown in *Figure 5AB*. The effects remained positive for all immune mediators in this subset analysis, indicating that the differences we saw between pre- and post-first sex samples were not solely a result of acquisition of STIs or pregnancy.

In addition, we sought to assess whether the differences between pre- and post-first sex would be explained by social determinants of health. There was limited diversity in our cohort in these variables, in particular in terms of race and ethnicity. We adjusted the models from our primary analysis for age, education, whether the participants reported a regular source of income, whether the participants

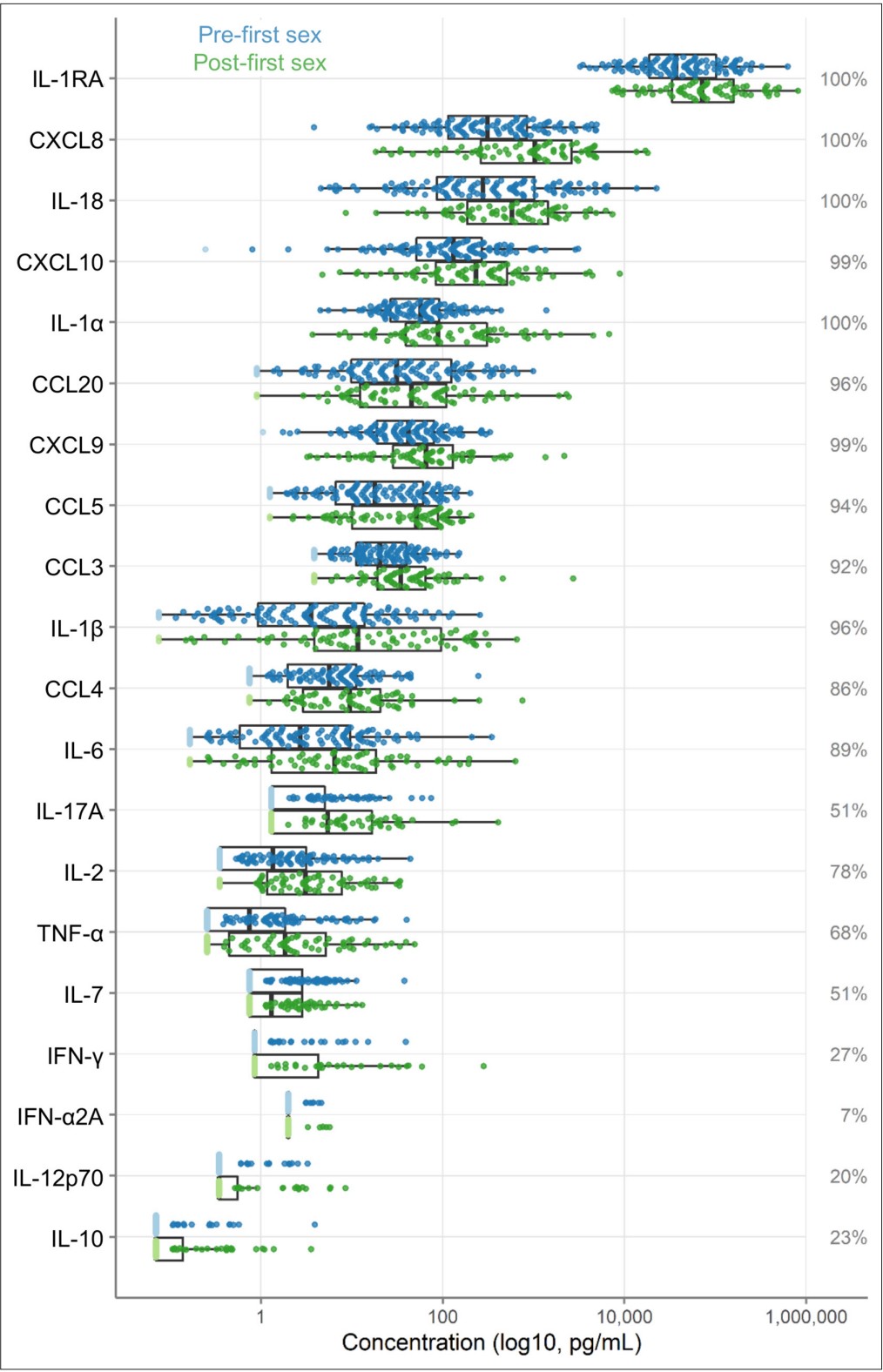

**Figure 2.** Immune factor concentrations in pg/mL from cervicovaginal lavage specimens from adolescent girls and young women, comparing specimens from before and after first sexual intercourse. Blue indicates specimens prior to first sex, and green indicates samples after first sex. Specimens below the limit of detection are indicated as lighter colors and were set to half the limit of detection. Percentages indicate the percent of samples within the detectable range of the assays. The boxes range from the 25th to 75th percentiles of the data, with the middle

*Figure 2 continued on next page*

*Figure 2 continued*

vertical line indicating the median. The whiskers stretch to the values no farther from the edge of the box than 1.5 times the interquartile range. The sample size is 180 specimens from 95 participants.

lived in an urban or rural environment, and whether the participants lived in a home with a metal or tile roof (lacking such a roof being an indicator of poverty). As shown in *Figure 5AB*, the effect sizes from the multivariate models were similar to those from the univariate models, suggesting that these variables do not explain the results we observed.

To determine whether the pre- and post-first sex differences were driven by differences in protein recovery, we normalized the immune mediator concentrations to total protein concentrations. This analysis modestly reduced the effect size estimates between pre- and post-first sex (*Figure 5AB*). This result is consistent with the slightly higher total protein concentrations observed in post-first sex CVLs (*Figure 5C*, 0.25 log2 fold change mg/mL, p=0.12). In all cases except for IL-7, the immune mediator concentrations remained higher post-first sex after normalization to total protein.

## Fully paired analysis

As described above, only a minority of participants provided both pre- and post-first sex samples; about two-thirds of participants provided only pre-first sex or only post-first sex samples. This is a consequence of the difficulty of assessing participants during dynamic life events and is consistent with prior studies (*Ghosh et al., 2018*; *Jespers et al., 2016*).

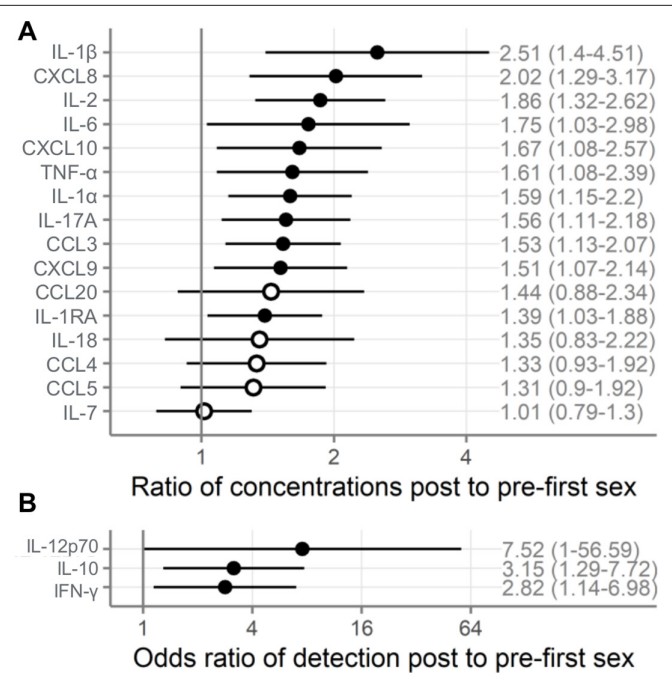

**Figure 3.** Comparison of immune mediators in cervicovaginal lavage samples pre- and post-first sexual intercourse. Univariate mixed-effect models with first sex as fixed effect and participant as random effects. Symbols indicate the mean and horizontal lines indicate the 95% confidence intervals. Filled symbols indicate p<0.05, while open symbols indicate p≥0.05. Vertical lines at 1 indicate no difference between pre- and post-first sex. (**A**) Multiplicative ratio of post-first sex concentrations (pg/mL) to pre-first sex concentrations on a log scale. A value of 1 indicates no difference, and numbers higher than 1 indicate greater quantities of immune mediators post-first sex. Ratios and 95% confidence intervals are shown at right. The ratio is calculated as 2^mean log2 difference shown in *Table 2*. ( **B**) Odds ratio (on a log scale) comparing the odds of the immune mediator being detected above the lower limit of detection (for immune mediators detected in fewer than half of the samples). A value of 1 indicates no difference, and numbers higher than 1 indicate greater detection of immune mediators post-first sex. Ratios and 95% confidence intervals are shown at right. The sample size is 180 specimens from 95 participants.

**Table 2.** Comparison of immune mediators in cervicovaginal lavage samples pre- and post-first sexual intercourse.

Results of univariate mixed-effect models with first sex as fixed effect and participant as random effects. Mean log2 pg/mL difference values above 0 indicate higher concentrations post-first sex. Ratio indicates the multiplicative ratio of post over pre-sex (on a linear scale), so a ratio of 1 indicates no difference, ratios >1 indicate higher post, and ratios <1 indicate higher pre. The ratio is calculated as 2^mean log2 difference. Adjusted p-values are adjusted by Holm-Bonferroni for 19 immune mediators.

| Immune mediator | Mean log2 difference | Standard error | Ratio (95% CI) | p-Value | Adjusted p-value |
|---|---|---|---|---|---|
| IL-2 | 0.90 | 0.25 | 1.86 (1.32–2.62) | 4.7E-4 | 0.009 |
| IL-1β | 1.33 | 0.43 | 2.51 (1.4–4.51) | 0.002 | 0.044 |
| CXCL8 | 1.01 | 0.33 | 2.02 (1.29–3.17) | 0.003 | 0.045 |
| IL-1α | 0.67 | 0.24 | 1.59 (1.15–2.2) | 0.006 | 0.088 |
| CCL3 | 0.62 | 0.22 | 1.53 (1.13–2.07) | 0.006 | 0.091 |
| IL-17A | 0.64 | 0.25 | 1.56 (1.11–2.18) | 0.011 | 0.154 |
| TNF-α | 0.69 | 0.29 | 1.61 (1.08–2.39) | 0.020 | 0.236 |
| CXCL10 | 0.74 | 0.32 | 1.67 (1.08–2.57) | 0.021 | 0.236 |
| CXCL9 | 0.60 | 0.26 | 1.51 (1.07–2.14) | 0.021 | 0.236 |
| IL-1RA | 0.48 | 0.22 | 1.39 (1.03–1.88) | 0.032 | 0.258 |
| IL-6 | 0.81 | 0.39 | 1.75 (1.03–2.98) | 0.041 | 0.284 |
| CCL4 | 0.42 | 0.27 | 1.33 (0.93–1.92) | 0.124 | 0.621 |
| CCL20 | 0.53 | 0.36 | 1.44 (0.88–2.34) | 0.146 | 0.621 |
| CCL5 | 0.39 | 0.28 | 1.31 (0.9–1.92) | 0.166 | 0.621 |
| IL-18 | 0.44 | 0.36 | 1.35 (0.83–2.22) | 0.232 | 0.621 |
| IL-7 | 0.02 | 0.18 | 1.01 (0.79–1.3) | 0.921 | 0.921 |

We repeated our univariate analysis on the 66 samples from the 33 participants who provided paired pre- and post-first sex samples. As shown in *Figure 5AB*, immune mediator concentrations were generally higher post-first sex in this subset of samples, but the effects were smaller and all but one (IL-12p70) failed to reach statistical significance.

The specimens from the fully paired subset were generally comparable to the specimens from the participants who only provided post-specimens, with one important difference: the post-specimens in the fully paired subset were obtained sooner after reported date of first sex (p=4.7E-8). In the fully

**Table 3.** Comparison of immune mediators in cervicovaginal lavage samples pre- and post-first sexual intercourse.

Results of univariate mixed-effect models with first sex as fixed effect and participant as random effects. Log odds ratio values above 0 indicate that the immune mediator was more often detected post-first sex. Odds ratio indicates the odds ratio of detection for post over pre-sex (on a linear scale), so a ratio of 1 indicates no difference, ratios >1 indicate higher detection post, and ratios <1 indicate higher detection pre. Adjusted p-values are adjusted by Holm-Bonferroni for 19 immune mediators.

| Immune mediator | Log-odds | Standard error | Odds ratio (95% CI) | p-Value | Adjusted p-value |
|---|---|---|---|---|---|
| IL-10 | 1.15 | 0.46 | 3.15 (1.29–7.72) | 0.012 | 0.154 |
| IFN-γ | 1.04 | 0.46 | 2.82 (1.14–6.98) | 0.025 | 0.236 |
| IL-12p70 | 2.02 | 1.03 | 7.52 (1.00–56.59) | 0.050 | 0.300 |

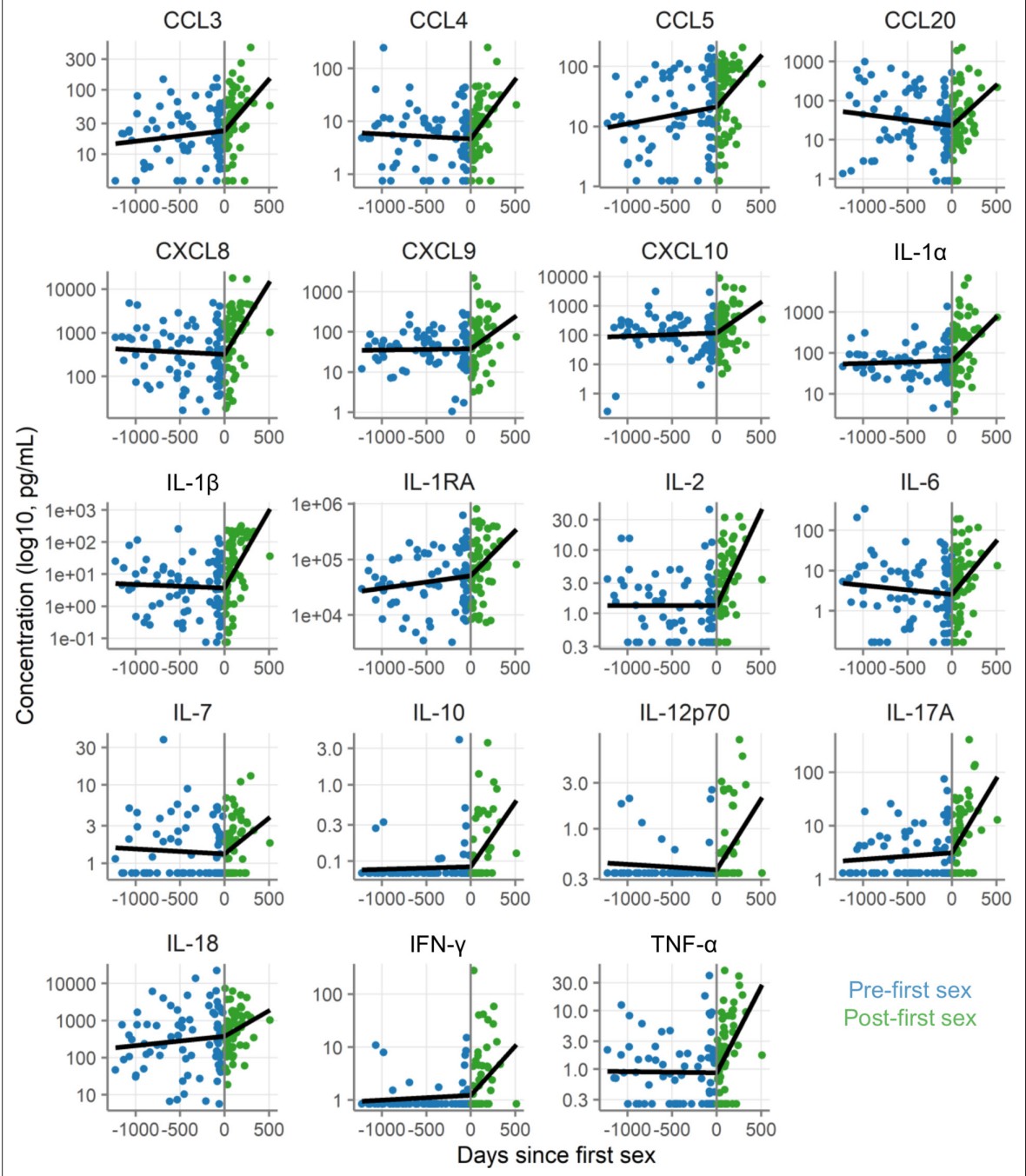

**Figure 4.** Concentration of immune mediators in cervicovaginal lavage samples relative to day of first sex. Each symbol depicts the concentration in a single sample. Black lines show the slopes from mixed-effect models with days since first sex as fixed effect and participant as random effects with linear splines using a knot at day 0. Concentrations are shown in pg/mL on a log10 scale. The sample size is 80 pre-first sex samples from 59 participants and 60 post-first sex samples from 49 participants.

paired subset, the post-first sex samples came only about a month after first sex (median 31.5 days). In contrast, the post-first sex samples from the other participants came about 4 months after first sex (median 157 days). The age of the participants at the time of first sex, however, was similar (18.9 years in each subset, p=0.90).

**Table 4.** Change in concentration of immune mediators with time.

Results of mixed-effect models with 100 days since first sex as a fixed effect and participant as random effects, using a spline with a knot at 0 (the day of first sexual intercourse). The slope column indicates the change in log2 immune mediator concentration (pg/mL) per 100 days. A value of 1 would indicate an increase of 1 log2 pg/mL in 100 days, while a value of –1 would indicate a decrease of the same. The p-values test whether the post-first sex slope differs from the pre-first sex slope.

| | Slope pre-first sex | | Slope post-first sex | | p-Value for difference between slopes |
|---|---|---|---|---|---|
| | *Slope* | *95% CI* | *Slope* | *95% CI* | |
| IL-2 | 0.0003 | (–0.1, 0.1) | 0.99 | (0.59, 1.39) | 0.00003 |
| IL-12p70 | –0.02 | (–0.07, 0.03) | 0.49 | (0.28, 0.69) | 0.00003 |
| IL-1β | –0.04 | (–0.21, 0.13) | 1.60 | (0.92, 2.29) | 0.0001 |
| IL-10 | 0.01 | (–0.05, 0.07) | 0.57 | (0.32, 0.81) | 0.0002 |
| CXCL8 | –0.03 | (–0.16, 0.09) | 1.09 | (0.57, 1.61) | 0.0002 |
| TNF-α | –0.01 | (–0.12, 0.1) | 0.98 | (0.52, 1.44) | 0.0002 |
| IL-17A | 0.04 | (–0.06, 0.14) | 0.93 | (0.51, 1.34) | 0.0003 |
| CCL4 | –0.03 | (–0.13, 0.07) | 0.75 | (0.33, 1.17) | 0.001 |
| IFN-γ | 0.03 | (–0.05, 0.11) | 0.62 | (0.28, 0.97) | 0.003 |
| IL-1α | 0.02 | (–0.09, 0.13) | 0.71 | (0.29, 1.13) | 0.005 |
| IL-6 | –0.08 | (–0.24, 0.08) | 0.88 | (0.24, 1.53) | 0.01 |
| CCL3 | 0.05 | (–0.03, 0.13) | 0.53 | (0.2, 0.87) | 0.01 |
| CCL20 | –0.10 | (–0.25, 0.06) | 0.69 | (0.07, 1.3) | 0.03 |
| CXCL9 | 0.01 | (–0.1, 0.12) | 0.54 | (0.1, 0.97) | 0.03 |
| CXCL10 | 0.03 | (–0.11, 0.18) | 0.70 | (0.14, 1.26) | 0.04 |
| IL-1RA | 0.07 | (–0.03, 0.18) | 0.54 | (0.14, 0.94) | 0.04 |
| IL-7 | –0.02 | (–0.09, 0.05) | 0.31 | (0.02, 0.59) | 0.04 |
| CCL5 | 0.09 | (–0.03, 0.22) | 0.56 | (0.08, 1.04) | 0.09 |
| IL-18 | 0.08 | (–0.06, 0.22) | 0.46 | (–0.12, 1.04) | 0.25 |

## Systematic review and meta-analysis of immune mediator changes post-first sex

We performed a systematic review to identify comparable studies. Of 147 abstracts retrieved through our search, 7 were assessed as potentially eligible, and 2 studies were determined to be eligible after review of the full texts. We obtained individual participant data from both studies. The first was a cross-sectional study by ELISA of 11–19-year olds in Washington DC with 18 CVL samples from 10 participants post-first sex and 8 participants pre-first sex. This study found a statistically significant decrease in TNF-α, while most other mediators were higher (but not statistically significant) post-first sex (*Ghosh et al., 2018*). The other study longitudinally followed 14–19-year-old AGYW in Belgium, with 269 swab samples from 93 participants, 9 of whom provided both pre- and post-first sex samples, 43 of whom provided only pre-samples, and 41 of whom provided only post-samples. Using Luminex, this study found increases in IL-1α, IL-1β, and CXCL8 post-first sex (*Jespers et al., 2016*). No important issues were identified in checking the integrity of the individual participant data received for these two studies. Risk of bias was low (scores of 7/7 for *Jespers et al., 2016* and the study reported here) to moderate (score of 4/7 for *Ghosh et al., 2018*). Individual participant data (IPD) was obtained from all eligible studies, so there is no additional risk of bias from missing data.

Including our study reported here, nine immune mediators were measured in at least two studies and were eligible for meta-analysis (*Figure 6*; *Table 5*). Meta-analyses identified six immune mediators

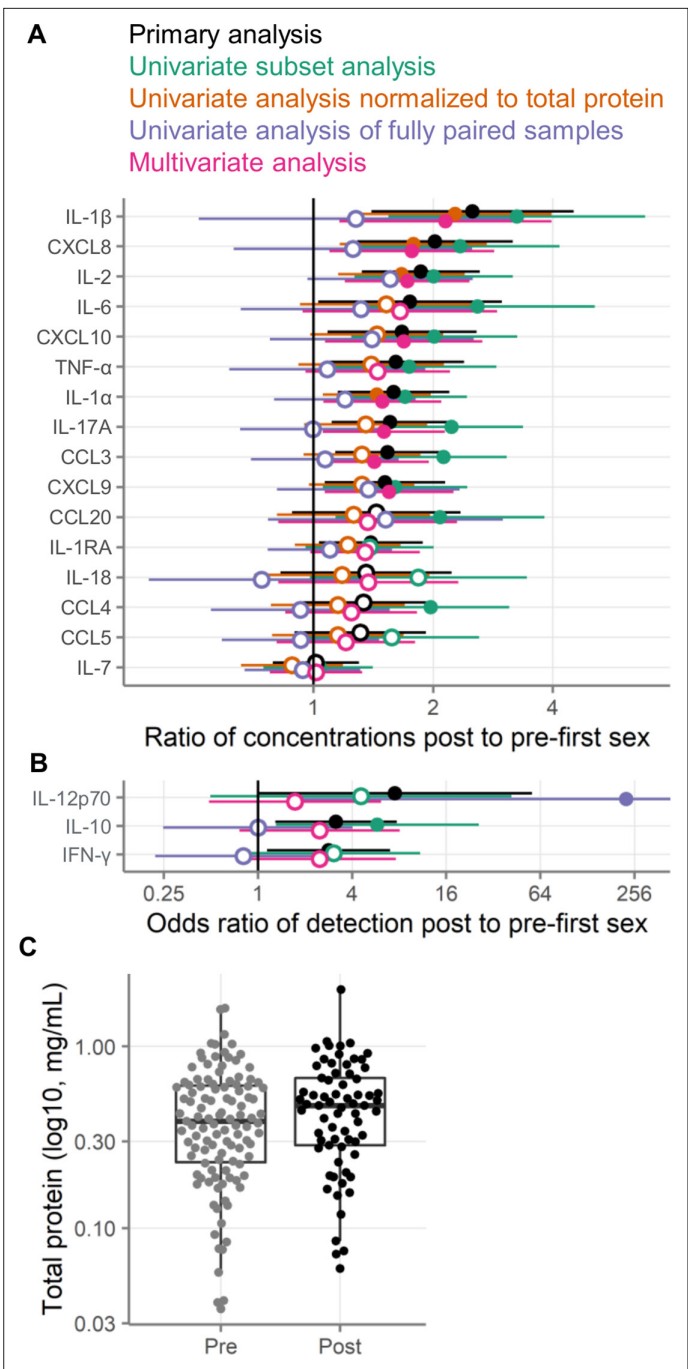

**Figure 5.** Alternative analysis strategies for the association of first sexual intercourse and quantities of cervicovaginal immune mediators. Filled symbols indicate p-value <0.05, and open symbols indicate p-values ≥0.05. Vertical lines at 1 indicate no difference between pre- and post-first sex. Black symbols show the same primary analysis as *Figure 3*. Green symbols show the primary, univariate models applied to a subset of samples negative for pregnancy, contraception, bacterial vaginosis (BV), chlamydia, and HSV-2. Orange symbols show the primary univariate analysis performed on concentrations normalized to total protein concentrations (pg/mg protein). Purple symbols show the primary analysis repeated on only those samples from participants who provided both pre- and post-first sex samples. Pink symbols show a multivariate analysis adjusted for age, education, income, urban/rural residence, and roof type. (**A**) Multiplicative ratio of post-first sex concentrations to pre-first sex concentrations (pg/mL) on a log scale. A value of 1 indicates no difference, and numbers greater than 1 indicate higher values post-sex. (**B**) Odds ratio (on a log scale) for the immune mediator being detected above the lower limit of detection. A value of 1 indicates no difference, and numbers greater than 1 indicate higher odds

*Figure 5 continued on next page*

*Figure 5 continued*

of detection post-sex. Symbols indicate the mean, and horizontal lines indicate the 95% confidence intervals. The error bar for the fully paired analysis of IL-12 extends off-scale. (**C**) Total protein concentrations in cervicovaginal lavage specimens. The boxes range from the 25th to 75th percentiles of the data, with the middle horizontal line indicating the median. The whiskers stretch to the values no farther from the edge of the box than 1.5 times the interquartile range. The sample size is 180 specimens from 95 participants.

as having higher concentrations post-first sex (IL-1α, IL-1β, IL-6, CXCL8, CCL4, and CCL5, all p<0.05; all p<0.05 after adjustment for multiple comparisons except for IL-6 and CCL4 [adjusted p=0.066]). The remaining three immune mediators all had meta-analysis p>0.05, with two higher post-first sex (CXCL10 and CCL20) and TNF-α showing the opposite. ***Supplementary file 4*** contains detailed plots for each immune mediator included in the meta-analyses: the concentration data from every sample for each study and a forest plot showing each study's effect size and weighting for each immune mediator.

We used statistical heterogeneity ($I^2$) to assess the comparability of the three studies included in the meta-analysis, with an $I^2$ of 0 indicating that all between-study variation can be explained by random sampling and higher $I^2$ proportions indicating clinical, biological, or methodological diversity between studies (***Higgins et al., 2021***). In our meta-analysis, statistical heterogeneity was generally low ($I^2$ of 0 for IL-1β, IL-6, CXCL8, and CCL4) to moderate ($I^2$ of 0.05–0.55 for IL-1α, CCL5, CXCL10, and CCL20; ***Table 5***). TNF-α had high statistical heterogeneity between studies ($I^2$=0.885).

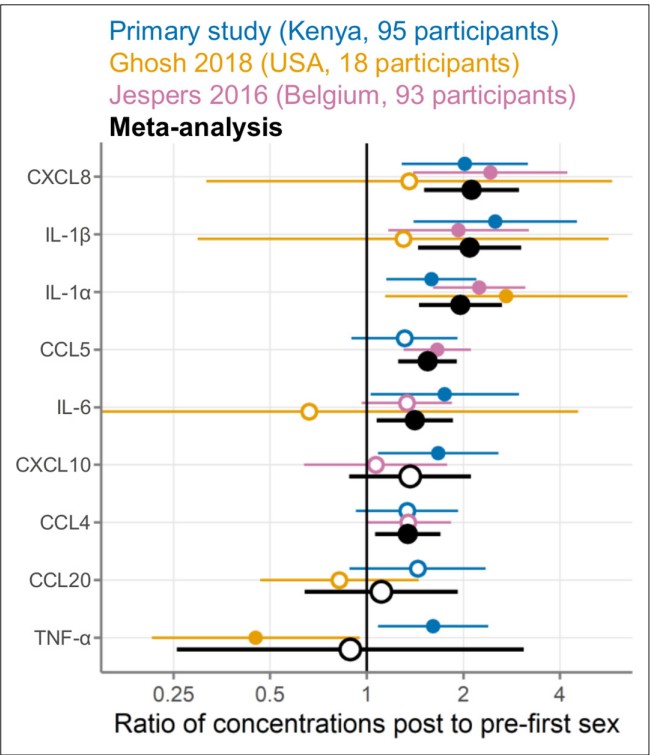

**Figure 6.** Meta-analysis comparing three studies measuring cervicovaginal immune mediators in adolescent girls and young women before and after first sexual intercourse. Multiplicative ratio of post-first sex concentrations to pre-first sex concentrations (pg/mL) on a log scale. A value of 1 indicates no difference, and numbers greater than 1 indicate higher values post-first sex. Symbols indicate the mean, and horizontal lines indicate the 95% confidence intervals. Filled symbols indicate p-value <0.05 while open symbols indicate p-values ≥0.05. Vertical lines at 1 indicate no difference between pre- and post-first sex. Colors indicate the source of the data (blue, primary study; gold, Ghosh 2018; pink, Jespers 2016; black, meta-analysis).

**Table 5.** Meta-analyses of immune mediator concentrations in cervicovaginal samples comparing pre- and post-first sexual intercourse.

Results of random effects meta-analyses using inverse variance pooling. Mean log2 difference values above 0 indicate higher concentrations post-first sex. Ratio indicates the multiplicative ratio of post over pre-sex (on a linear scale), so a ratio of 1 indicates no difference, ratios >1 indicate higher post, and ratios <1 indicate higher pre. Adjusted p-values are adjusted by Holm-Bonferroni for measuring nine immune mediators. $I^2$ is a measure of heterogeneity between studies, with a proportion of 0 indicating that all between-study variation can be explained by random sampling and higher $I^2$ proportions values indicating clinical, biological, or methodological diversity between studies.

| Immune mediator | Mean log2 difference | Standard error | Ratio (95% CI) | p-Value | Adjusted p-value | Number of studies | $I^2$ |
|---|---|---|---|---|---|---|---|
| IL-1α | 0.97 | 0.22 | 1.96 (1.45–2.64) | 1.0E-5 | 9.1E-5 | 3 | 0.28 |
| CXCL8 | 1.08 | 0.25 | 2.12 (1.51–2.98) | 1.5E-5 | 1.2E-4 | 3 | 0 |
| CCL5 | 0.63 | 0.16 | 1.55 (1.25–1.91) | 5.2E-5 | 3.6E-4 | 2 | 0.05 |
| IL-1β | 1.06 | 0.27 | 2.09 (1.44–3.03) | 9.3E-5 | 5.6E-4 | 3 | 0 |
| IL-6 | 0.50 | 0.20 | 1.41 (1.07–1.86) | 0.013 | 0.066 | 3 | 0 |
| CCL4 | 0.42 | 0.17 | 1.34 (1.06–1.7) | 0.014 | 0.066 | 2 | 0 |
| CXCL10 | 0.45 | 0.32 | 1.36 (0.88–2.11) | 0.164 | 0.492 | 2 | 0.42 |
| CCL20 | 0.15 | 0.40 | 1.11 (0.64–1.92) | 0.712 | 1 | 2 | 0.54 |
| TNF-α | –0.17 | 0.92 | 0.89 (0.26–3.08) | 0.851 | 1 | 2 | 0.89 |

## Discussion

In this study of a large cohort of Kenyan AGYW with specimens collected pre- and post-first sex, we observed an association of increased levels of CVT immune mediators with the start of sexual activity. The strongest evidence was for IL-1β, IL-2, and CXCL8, but all measured immune mediators followed a similar pattern. In addition, we observed strong linear associations between time since first sex and immune mediator concentration, compared to flat levels over time pre-first sex. These results were tested with several analytical approaches: limiting to only paired samples, limiting to samples without known causes of inflammation (BV, STIs, or pregnancy), adjusting for socioeconomic cofactors, and adjusting for multiple comparisons, but with all of these methods, the increased levels of immune mediators post-sex remained consistent.

We further performed a systematic review and meta-analysis of individual participant data, identifying two previous studies conducted in Belgium and the United States (*Jespers et al., 2016*; *Ghosh et al., 2018*). Meta-analysis combining all three studies found that the start of sexual activity was associated with increased concentrations for 8/9 immune mediators, with particularly strong evidence for IL-1α, IL-1β, IL-6, and CXCL8. The impressive agreement across these three studies strengthens the conclusion that the start of sexual activity is associated with increases in a wide range of CVT immune mediators.

While we observed a clear association between sexual activity and higher immune mediator concentrations, there is room for caution about whether the association is causal. For example, there could be differences in social determinants of health between the participants who do and do not engage in sexual activity. These differences were largely unmeasured in all three studies and may contribute to the changes we observed in immune mediator levels pre- and post-first sex. In our Kenyan study, we had information from the baseline visit about participant age, education, income, and housing. The effects of first sex on immune mediators remained the same after adjusting for these variables, suggesting that age, education, income, and housing at baseline do not explain the differences we observed. However, we did not assess these variables for changes over time, only at baseline.

The fully paired subset of our cohort (where participants provided pre- and post-samples) had smaller effect sizes than the full cohort. Most effects remained positive, but they were smaller in magnitude than in the primary analysis, and all were non-significant. A possible explanation for the smaller effects is that immune mediator concentrations increase over time post-first sex (*Figure 4*).

The post-first sex samples in the fully paired subset were collected sooner after first sex (median 32 days) than those from the participants who provided only post-first sex samples (median 157 days). Thus, the effects may have been weaker in the fully paired subset, in part, because the samples were obtained such a short time after first sex. Another possible explanation for the smaller effects is unmeasured confounding in the full cohort. Because participants in this subset provided both pre- and post-first sex samples, the subset largely controls for unmeasured confounding factors. These unmeasured factors could have inflated the effect sizes in the full cohort. A future study with a larger number of participants followed from pre- to post-first sex and with additional specimens collected for a longer period of time post-sex would be necessary to resolve this uncertainty.

While it has long been suspected that sexual activity induces CVT changes, our study, with post-first sex samples collected a median of only 66 days after first sex, clarifies that the association of inflammatory changes with sexual activity occurs very early after the start of sexual activity. In addition to an early start of the inflammatory changes post-first sex, we observed an accumulation over time. The rates of increase over time varied between immune mediators, but on average indicated a doubling in concentration within about 5 months. For example, the immune mediator with the median rate of increase was CCL20 and it increased by 0.69 log2 pg/mL per 100 days (*Figure 4*). Furthermore, the association appears even without any incipient BV or STI, and might need to be considered a normal component of sexual activity.

The observed increases in inflammation may be important clinically in modifying risk of STI acquisition and in promoting fertility. There has been speculation that AGYW are at increased risk of STI due to physiological differences in the CVT compared to older women, in addition to behavioral risk factors (*Hwang et al., 2011*; *Masson et al., 2015*; *McKinnon and Karim, 2016*; *Venkatesh and Cu-Uvin, 2013*). Our research shows that there is an association between starting to have sex and increases in immune mediator concentration. Whether the increase in immune mediators brings protection from or vulnerability to STIs is unclear. Increased inflammation may prepare the CVT to prevent infection upon exposure. At the same time, increased expression of inflammatory mediators may recruit CD4 T cells, increasing the abundance of HIV target cells and potentially the risk of HIV infection. In addition to preventing infection, immune mediators play important roles in implantation and pregnancy. Therefore, the associations we observed with increases in immune mediators and start of sexual activity may be relevant for fertility. For example, IL-6 induces sperm capacitation, increasing its fertilizing ability (*Laflamme et al., 2005*). In addition, trophoblast cells secrete immune mediators to attract and regulate immune cells within the placenta, which is a necessary process for successful pregnancy (*Mor et al., 2011*).

## Mechanism of immune mediator increase

Several mechanisms may play a role in the association of sexual activity with increased CVT immune mediator concentrations: BV, semen exposure, vaginal practices including washing, and physical microtrauma.

Numerous studies show that BV is rare prior to first sex, increases after first sex (*Tabrizi et al., 2006*; *Fethers et al., 2012*; *Fethers et al., 2011*; *Fethers et al., 2009*; *Francis et al., 2020*; *Francis et al., 2019*; *Mitchell et al., 2012*; *Thoma et al., 2011a*), is associated with recent sexual activity, and is associated with exposure to semen (*Gajer et al., 2012*; *Ma et al., 2013*; *Jewanraj et al., 2021*; *Mngomezulu et al., 2021*; *Thoma et al., 2011b*). Therefore, vaginal bacteria may play a role in these increased immune mediator levels. Jespers et al. reported that the increased CVT cytokines they observed post-sex were partly mediated by increases in detection and concentration of *Gardnerella vaginalis* and *Atopobium vaginae* (*Jespers et al., 2016*). Our finding of higher IL-1α and IL-1β post-first sex is consistent with literature linking BV to increased CVT IL-1α and IL-1β. However, BV is also associated with reduced CXCL10 (*Masson et al., 2019*), which conflicts with our finding of increased CXCL10 after first sex. Our data demonstrate conclusively that even when limited to only women with no BV, immune mediators were increased post-first sex. However, a leading hypothesis for increased CVT concentrations post-first sex may be the establishment of non-Lactobacillus bacterial populations.

Exposure to semen has been associated with increases in cervical leukocytes (*Pandya and Cohen, 1985*; *Thompson et al., 1992*; *Sharkey et al., 2012*) and inflammatory cytokines, especially IL-6 and CXCL10, in most (*Jewanraj et al., 2021*; *Mngomezulu et al., 2021*; *Sharkey et al., 2012*; *Francis et al., 2016*; *Jespers et al., 2017*; *Kyongo et al., 2012*) but not all studies (*Agnew et al., 2008*;

*Nakra et al., 2016*). In vitro exposure to seminal plasma induces cytokines including IL-1α, IL-6, and CXCL8 in cervical and vaginal epithelial cells (*Sharkey et al., 2007*) and cervical explants (*Adefuye et al., 2014*; *Denison et al., 1999*; *Introini et al., 2017*). Others have suggested that increases in immune mediators following exposure to semen may be a result of semen directly transferring cytokines into the CVT (*Deese et al., 2021*). However, detection of immune mediators directly transferred by semen is unlikely to explain our results, because semen was rarely detected in our study (by Y-chromosome DNA and PSA).

Vaginal washing changes the microbiome and increases BV (*Lokken et al., 2019*; *Low et al., 2011*). In this cohort, vaginal washing was reported by about 56% of participants, but data was not available for vaginal washing for the particular specimens analyzed here, and data was not available for how vaginal washing practices may have changed after first sex.

Penile-vaginal sexual intercourse can cause microtrauma in the vaginal walls (*Norvell et al., 1984*), and this process may be inflammatory. However, exposure to semen may be necessary for the inflammatory response to intercourse, because condoms have been shown to block that response (*Sharkey et al., 2012*). Our study was unable to evaluate the role of microtrauma in the inflammatory response observed.

## Strengths and limitations

The strengths of our study include a large, well-characterized cohort with multiple specimens per participant. Our sampling technique allowed collection of CVL without a speculum exam, which improved the feasibility of this research in a population with limited sexual experience. In addition, our systematic review and meta-analysis place our results in the context of what is known in the literature and synthesize all three studies. Despite the diversity of these studies and populations, all three studies showed consistent results.

An important limitation of our study is possible misclassification of specimens from pre-sexual activity. It is important to note that all three cohorts presented here were specifically designed to assess the sexual activity of adolescents and used best practices in ascertaining this challenging key variable. Of the 123 samples where participants reported no sexual activity, we classified 12 (9.8%) as post-first sex based on STI, pregnancy, Y-chromosome, or PSA results. Misclassification of specimens as pre/post-first sex is a challenge in this field, which we did our best to mitigate, but residual misclassification is possible.

In a secondary analysis, we removed all samples where the participant used contraception, was pregnant, had a Nugent score above 3, or was positive for HSV-2 or CT. The results of this analysis were similar to our primary, univariate analysis of the full cohort. These factors have complex relationships with sexual activity: Each can be caused by sexual activity and several of these variables could also be influenced (in part) by changes in immune mediator level. Thus, they may mediate the effect of sex on immune mediator levels, or they may be a consequence of it. Our goal with this secondary analysis was not to precisely estimate the direct effect of beginning sexual activity in the absence of these factors, but rather to determine whether these factors solely explained our results. Because the effects remained after removing these samples, we conclude that these factors by themselves do not solely explain the results we observed.

We observed similar results across three different studies in both low- and high-income settings. The largest limitation is that the total sample size across all three studies was less than 500 AGYW. Our study did include a number of cases of chlamydia and BV, increasing its generalizability, given the commonness of these conditions. However, our study notably included few cases of HSV-2, a very common STI, which may reduce its generalizability if HSV-2 modifies the effect of sexual activity on immune mediators. Similarly, we excluded a small number of samples due to trichomonas or gonorrhea infection or active local HSV-1/HSV-2 production because there were too few samples to reliably analyze. Moreover, our Kenyan cohort included little socioeconomic diversity, with participants being similar in income, housing, education, and ethnicity but still revealed similar results to AGYW on other continents.

We observed higher total protein concentrations in the post-first sex samples, translating to smaller differences between pre- and post-first sex groups when immune mediator concentrations were normalized to total protein. It is possible that the increases in total protein concentration post-first sex are associated with the general increases in immune mediators observed post-first sex. However,

immune mediators remained higher post-first sex even after normalizing to total protein, so global protein upregulation does not completely explain this phenomenon.

The three studies included in the meta-analysis differed in important ways which could affect comparability. These differences included country of origin, presence of BV in the population, sample collection method (*Dezzutti et al., 2011*), and immunoassay platform (*Chowdhury et al., 2009*). However, statistical heterogeneity was low to moderate for most immune mediators, indicating that the between-study variability was largely what would be expected due to random sampling. The high statistical heterogeneity for TNF-α indicates a need for further study of that mediator. Because we obtained the raw individual participant data from each study, we were able to analyze all three studies using the same methods, which improves their comparability.

## Conclusions

We identified consistent increases in cervicovaginal immune mediators among participants after first penile-vaginal sexual intercourse. The dynamic nature of the inflammatory milieu after sexual activity may be a catalyst for changes that could promote acquisition of STIs. Because these specimens are from AGYW with little pre-existing BV or STI, our research is suggestive that this inflammatory milieu may provide risk independent of STIs or vaginal dysbiosis. Further research should focus on the exact causes of inflammation associated with first sex and assess whether this inflammation is potentially harmful or whether it offers any benefits. We envision that this information will contribute to expanding our toolset in the fight against the high STI susceptibility observed in AGYW.

## Acknowledgements

The authors would like to thank the following for their contributions: study staff and participants in Thika, Kenya; the Endocrine Technologies Core (NIH P51OD011092) at the Oregon National Primate Research Center for measuring progesterone concentrations; Dr Scott McClelland and the University of Washington/University of Nairobi East Africa STI Laboratory in Mombasa, Kenya for STI testing; Diana Louden from the University of Washington Health Sciences Library for devising the systematic review search strategy; Dr. Veronica Gomez-Lobo for performing the clinical components of the Washington DC study. Funding: This research was funded by R01 HD091996-01 (ACR), by P01 AI 030731–25 (Project 1) (AW), R01 AI116292 (FH), R03 AI154366 (FH), and by the Center for AIDS Research (CFAR) of the University of Washington/Fred Hutchinson Cancer Research Center AI027757. Laboratory work was done at the CFAR Core Immunology Laboratory, supported by AI027757. The funders had no role in study design, data collection and analysis, or preparation of the manuscript. The content is solely the responsibility of the authors and does not necessarily represent the official views of the National Institutes of Health.

## Additional information

### Competing interests

Kenneth Ngure: Kenneth Ngure was supported by the International AIDS Society to attend AIDS 2022. The author has no other competing interests to declare. Nelly R Mugo: Nelly R Mugo received honoraria from MERCK Ltd in support of a presentation on HPV vaccination uptake in LMIC. The author has no other competing interests to declare. The other authors declare that no competing interests exist.

### Funding

| Funder | Grant reference number | Author |
|---|---|---|
| Eunice Kennedy Shriver National Institute of Child Health and Human Development | HD091996 | Alison C Roxby |
| National Institute of Allergy and Infectious Diseases | AI030731 | Anna Wald |

| Funder | Grant reference number | Author |
|---|---|---|
| National Institute of Allergy and Infectious Diseases | AI116292 | Florian Hladik |
| National Institute of Allergy and Infectious Diseases | AI154366 | Florian Hladik |
| National Institute of Allergy and Infectious Diseases | AI027757 | Florian Hladik |

The funders had no role in study design, data collection and interpretation, or the decision to submit the work for publication.

## Author contributions

Sean M Hughes, Conceptualization, Data curation, Formal analysis, Visualization, Methodology, Writing – original draft; Claire N Levy, Katie A Martinez, Data curation, Investigation, Writing – review and editing; Fernanda L Calienes, Data curation, Investigation; Stacy Selke, Data curation, Investigation, Methodology, Project administration, Writing – review and editing; Kenneth Tapia, Formal analysis, Methodology, Writing – review and editing; Bhavna H Chohan, Lynda Oluoch, Catherine Kiptinness, Resources, Project administration, Writing – review and editing; Anna Wald, Conceptualization, Supervision, Funding acquisition, Writing – review and editing; Mimi Ghosh, Liselotte Hardy, Resources, Writing – review and editing; Kenneth Ngure, Nelly R Mugo, Conceptualization, Resources, Funding acquisition, Project administration, Writing – review and editing; Florian Hladik, Alison C Roxby, Conceptualization, Supervision, Funding acquisition, Methodology, Writing – original draft, Project administration

## Author ORCIDs

Sean M Hughes http://orcid.org/0000-0002-9409-9405
Claire N Levy http://orcid.org/0000-0003-3204-211X
Katie A Martinez http://orcid.org/0000-0002-3411-065X
Bhavna H Chohan http://orcid.org/0000-0002-2125-6799
Anna Wald http://orcid.org/0000-0003-3486-6438
Liselotte Hardy http://orcid.org/0000-0002-8785-5081
Florian Hladik http://orcid.org/0000-0002-0375-2764
Alison C Roxby http://orcid.org/0000-0001-9449-8062

## Ethics

Human subjects approval was obtained from the Kenya Medical Research Institute Scientific Ethics Review Unit (protocol 2760) and the University of Washington Institutional Review Board (number 00000946). Participants under age 18 provided written informed assent, and written informed consent was obtained from a parent/guardian. Participants assented privately from parents/guardians, after asking questions and deciding whether they wanted to participate free of parental influence. Participants age 18 or older provided written informed consent. No personally identifiable information was obtained from the studies included in the meta-analysis. Participants in both studies provided informed consent and consent to publish as described in the original manuscripts.

## Decision letter and Author response

Decision letter https://doi.org/10.7554/eLife.78565.sa1
Author response https://doi.org/10.7554/eLife.78565.sa2

# Additional files

## Supplementary files

• Supplementary file 1. Results of analyses. The results of the main analyses reported in the paper, including complete effect sizes, measures of variance, and p-values.

• Supplementary file 2. Systematic review search terms. The search string used to query PubMed to identify potentially relevant articles for the systematic review.

• Supplementary file 3. Risk of bias assessment scale. The scale used to assess the risk of bias in studies included in the meta-analysis. This scale was custom modified from the Newcastle-Ottawa

Quality Assessment scale.

• Supplementary file 4. Plots of every meta-analysis separately. Raw concentration data plots and forest plots for every immune mediator included in the meta-analysis section. The concentration plots show the measured immune mediator concentration for every sample in every study on a log10 scale, with samples from the same individual connected by lines. The forest plots show the effect size and weight in each study as well as the overall meta-estimate.

• MDAR checklist

• Source code 1. R code and raw data to reproduce the analysis. An R project which cleans the raw data, runs all of the models reported in the paper, and generates the figures and tables.

• Reporting standard 1. The preferred reporting items for a systematic review and meta-analysis of individual participant data (PRISMA-IPD) checklist. The completed checklist indicating compliance with recommended PRISMA-IPD guidelines for performing a meta-analysis of individual participant data.

### Data availability
All raw data and analysis code are available in the supplementary files.

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
