## [Editor Report]

This study finds that the levels of many immune markers are higher in vaginal samples in women taken after initiation of vaginal sex than before initiation of vaginal sex. This result may indicate that initiation of vaginal sex potentially influences vaginal immune responses in adolescents and young adults. This study will be of the highest interest to those interested in how immune markers can change within individuals over time.

---

## [Decision Letter]

**Decision letter after peer review:**

Thank you for submitting your article "Cervicovaginal immune mediators increase when young women begin to have sexual intercourse: a prospective study and meta-analysis" for consideration by *eLife*. Your article has been reviewed by 2 peer reviewers, one of whom is a member of our Board of Reviewing Editors, and the evaluation has been overseen Ricardo Azziz as the Senior Editor. The reviewers have opted to remain anonymous.

Essential revisions:

1. Please consider and discuss the potential for unmeasured confounding and mediation in the analysis, with potentially changes to the regression models in response to reviewer comments.

2. Please temper causal claims in the abstract and Discussion and provide further discussion as to alternative potential explanations for the findings.

3. Provide further description in the methods of the spline analysis.

4. Discuss the consistency in effect sizes between the paired and unpaired analyses.

5. Provide further discussion on whether there was difficulty in obtaining pre-sex samples from women, and whether there is a potential for selection bias due to unrepresentative sampling prior to sex.

*Reviewer #1 (Recommendations for the authors):*

– The authors use causal language throughout the paper, implying that sex increases the level of immune markers. While they are welcome to hypothesize this in the discussion, I would suggest being more cautious when describing the results in the main text, abstract, and title and use more descriptive language instead (Cervicovaginal immune mediators are higher in vaginal samples of women who report having initiated sex, rather than "Cervicovaginal immune mediators increase when young women begin to have sexual intercourse" – increase is a verb that implies causality), given the observational nature of the study and the potential for bias. I would suggest discussing more of the possibility of the biases mentioned below.

– In Table 1 I would have liked to see more demographic and socioeconomic variables measured in the study. As mentioned, studies of sexual behavior are very vulnerable to confounding due to social determinants of health. While it is possible that these variables are not different in pre- and post- sex samples in this study, it is necessary to show this for the reader to assess whether there is a potential for confounding.

– The variables included in the multivariate model are not appropriate if the objective is to adjust for confounders, as many of the variables are not confounders and are rather mediators or potentially downstream effects of immune markers. Adjusting for these would lead to a biased estimate of the causal effect of vaginal sex. The authors should take more time to reflect on the potential causal relationships between these variables and how they influence their analysis. Some reflections:

– Age: is the most clear-cut confounder in this analysis and should certainly be included in the model, as age influences sexual behavior and is likely to be correlated with levels of immune markers.

– Menstrual phase: is appropriate to include in the model despite not necessarily meeting the traditional definition of a confounder. The menstrual phase is unlikely to influence the initiation of sex. However, there may be chance differences in menstrual phase between samples pre- and post-sex that may warrant adjustment. As the menstrual phase is not a mediator or a downstream effect of sex, adjustment for this variable will not bias results.

– Pregnancy and contraception: these variables are potential mediators on the causal pathway between initiation of sex and levels of immune markers, and should not be included in the model. If the objective of the authors is to measure only the direct (non-mediated) effect of sex, then a more complex mediation analysis would be needed to disentangle the effects of sex, pregnancy, and contraception, and the model would need to include potential confounders which are associated with pregnancy and contraception use (ex. socioeconomic variables).

– Nugent score, Chlamydia infection, and HSV-2 seropositivity: these are complex variables that could be both mediators of the effects of sex but also downstream effects of changes in levels of immune markers. It is therefore problematic to adjust for these as they are potentially colliders on the causal pathway, and adjustment would potentially lead to selection (collider) bias. See the article by Hernán (Hernán A Structural Approach to Selection Bias, Epidemiology: 2004 – Volume 15 – Issue 5 – p 615-625 https://doi.org/10.1097/01.ede.0000135174.63482.43), especially figures 3-4.

– Unmeasured confounders: there is potentially some residual unmeasured confounding from variables which influence sexual initiation and which also potentially influence immune response which are not included in the model. The most likely are variables related to social determinants of health (socioeconomic status, ethnicity, etc.). If the authors have access to these variables from the original study, they should consider their inclusion in the model, especially if they are strongly correlated with sexual initiation.

– The strongest evidence that there is likely to be potential selection bias and unmeasured confounding in the main (unpaired) analysis comes from the paired analysis, where the point estimates for the effect sizes are much smaller than in the main analysis. I think this analysis is the strongest analysis in the paper and am surprised in fact that this was not considered as the main analysis. The authors should discuss the results from this analysis more, as the paired analysis controls for many of the unmeasured confounders that would bias the results in the main analysis, so the results from this analysis are more likely to be valid estimates of causal effect than the results from the unpaired analysis.

– While the analysis for effect of time in Figure 4 is interesting, I would have liked to see p-values for the spline effect in the model. It is easy to fit different curves pre- and post-sex, but it does not necessarily follow that the addition of a spline leads to a significant improvement in the model performance. If the p-values for the spline are not significant, it would suggest the data does not support that there is an effect of time since initiating sex. Furthermore, this analysis should be described in the methods, I did not see this analysis described anywhere.

– The analysis of the correlation between estimates from different models (Figure 5D) has no analytical utility and should be removed. It is expected the results from different models would be correlated as they are based on the same data. High correlation between estimates from different analyses does not mean that the results or the model structure is valid; the validity of an analysis is based on subject matter knowledge and consideration of whether the analysis is appropriate for the question at hand. This analysis is therefore not informative.

– The logistic and linear regressions provide different information about the data; the logistic regression informs us on whether a particular marker is more likely to be detected in post-sex samples, whereas the linear regression informs us on whether the level of a marker is higher in post-sex samples. I would have therefore been interested in seeing the results from both regression models for all of the markers, instead of showing the linear regression results for some and the logistic regression results for others.

– I think Figure 3 is a good way to present results (multiplicative change in marker levels). I would suggest also adding the point estimates of the ratios and their confidence intervals on the right-hand side of each estimate (forest plot style), as the ratio estimates are more intuitive than the regression coefficients in Table 2. Also, the figures should distinguish between the scale of the axes (log 2) vs the units of the axes. Usually, the parentheses in the labels are used to indicate the axis units. The (log2) in the label may therefore confuse readers, as the axes do not show the effect of an increase in a log2 unit, but the ratios pre- vs. post-sex on a log2 scale. This could be better explained in the legend.

– Figure 5C and 5D appear to be mislabeled.

*Reviewer #2 (Recommendations for the authors):*

– How easy technically to get CVL from AGYW pre-first sex? How is it possible to discern the impact of the sampling on the mediators?

– Why exclude STIs if these can be controlled for in the MV models? Also unclear why differentiating sexual activity and STIs is so critical if the effects of both often co-occur after sexual activity is initiated. Perhaps this makes sense since STIs were rare, but it seems like it could affect generalizability.

– The sensitivity analysis of participants sampled only pre- and post- sex is a strength, as this comparison will likely reduce heterogeneity between individuals. Here the focus should be less on P-values and more on consistency between effect sizes. A correlation is presented, but this could be heterogeneous because of estimates being stronger or weaker in the paired analysis. Therefore it may make more sense to compare these on a case-by-case basis, depending on the consistency in these trends.

– Abstract- doesn't make sense to say a "median" 54% increase, as the % increase will mean something different for different markers, depending on their individual distributions. Markers with wider distributions will naturally have great % increases.

– Re the 57 samples reported after first sex, and the 12 samples deemed as such due to pregnancy, presence of semen biomarkers, STIs etc- is the assumption that the 12 samples did not overlap with the 57, e.g. represent under-reporting of first sex?

– In figure 5, even though most covariates didn't affect the estimates, would some of these be considered possible "mediators" as opposed to "confounders". E.g. BV has been associated with lower CXCL10 levels, so changes in BV post sex (which has also been shown) could lead to decreases in some of these key chemokines. This is worth considering, in particular since participants were sampled for the most part a long time before/after sex. It does appear in the Discussion. Is there clear enough evidence to conclude that the effects are unlikely to be due to sex alone? The Discussion of this on page 16 re: CXCL10 does not seem to match how the data are presented.

– Adjusting for protein concentration- certainly worth reporting this increases post-sex, but could this be due to general increases in inflammation following sex.

– Bottom of page 12, paragraph starting with "The specimens from the fully-paired…" – it is unclear what the point of this paragraph is.

---

## [Author Response]

Essential revisions:1. Please consider and discuss the potential for unmeasured confounding and mediation in the analysis, with potentially changes to the regression models in response to reviewer comments.

We have substantially changed the multivariate modeling approach and provided a fuller discussion of the potential for unmeasured confounding, as detailed in the response to reviewer 1 below.

2. Please temper causal claims in the abstract and Discussion and provide further discussion as to alternative potential explanations for the findings.

We have replaced language implying causality with language specifying associations throughout the Abstract and Discussion as well as in the title. We have also added two paragraphs near the beginning of the Discussion where we discuss alternative possible explanations for our findings. These changes are detailed in the response to reviewer 1 below.

3. Provide further description in the methods of the spline analysis.

To address this point, we added a new paragraph to the Methods and a new table to the Results section (new Table 4), which are further described in the response to reviewer 1 below.

4. Discuss the consistency in effect sizes between the paired and unpaired analyses.

We expanded our presentation of the differences between these two analyses and made it more nuanced. We moved it from the Strengths and Limitations section towards the end of the Discussion to feature more prominently close to the beginning. This change is detailed in the response to reviewer 1 below.

5. Provide further discussion on whether there was difficulty in obtaining pre-sex samples from women, and whether there is a potential for selection bias due to unrepresentative sampling prior to sex.

Our study attempted to recruit from as broad of a population as possible by using a community recruitment strategy (as opposed to recruiting from adolescents seeking clinical care). We collected CVL specimens without the use of a speculum exam, which made it more feasible to collect pre-sex specimens. In addition, we used the meta-analysis as a further way to verify results from outside of our own cohort to try to assess whether these results were applicable in other geographic populations and with different laboratories. We feel satisfied that our cohort results are similar to other cohorts in our meta-analysis. This issue is further discussed in the response to reviewer 2 below.

Reviewer #1 (Recommendations for the authors):– The authors use causal language throughout the paper, implying that sex increases the level of immune markers. While they are welcome to hypothesize this in the discussion, I would suggest being more cautious when describing the results in the main text, abstract, and title and use more descriptive language instead (Cervicovaginal immune mediators are higher in vaginal samples of women who report having initiated sex, rather than "Cervicovaginal immune mediators increase when young women begin to have sexual intercourse" – increase is a verb that implies causality), given the observational nature of the study and the potential for bias. I would suggest discussing more of the possibility of the biases mentioned below.

We have changed the language throughout describing the results as an association rather than suggesting causality. These changes are tracked in the revised manuscript; many were very small modifications of sentences, so we have not reproduced them here.

We added two paragraphs to the Discussion (pages 18-19) where we give a more nuanced assessment of the possibility of causality as well as alternative explanations for the results we observed. These paragraphs read as follows:

“While we observed a clear association between sexual activity and higher immune mediator concentrations, there is room for caution about whether the association is causal. For example, there could be differences in social determinants of health between the participants who do and do not engage in sexual activity. These differences were largely unmeasured in all three studies and may contribute to the changes we observed in immune mediator levels pre- and post-first sex. In our Kenyan study, we had information from the baseline visit about participant age, education, income, and housing. The effects of first sex on immune mediators remained the same after adjusting for these variables, suggesting that age, education, income, and housing at baseline do not explain the differences we observed. However, we did not assess these variables for changes over time, only at baseline.

“The fully paired subset of our cohort (where participants provided pre and post samples) had smaller effect sizes than the full cohort. Most effects remained positive, but they were smaller in magnitude than in the primary analysis and all were non-significant. A possible explanation for the smaller effects is that immune mediator concentrations increase over time post-first sex (Figure 4). The post-first sex samples in the fully paired subset were collected sooner after first sex (median 32 days) than those from the participants who provided only post-first sex samples (median 157 days). Thus, the effects may have been weaker in the fully paired subset, in part, because the samples were obtained such a short time after first sex. Another possible explanation for the smaller effects is unmeasured confounding in the full cohort. Because participants in this subset provided both pre- and post-first sex samples, the subset largely controls for unmeasured confounding factors. These unmeasured factors could have inflated the effect sizes in the full cohort. A future study with a larger number of participants followed from pre- to post-first sex and with additional specimens collected for a longer period of time post-sex would be necessary to resolve this uncertainty.”

We changed the title to “Starting to have sexual intercourse is associated with increases in cervicovaginal immune mediators in young women: a prospective study and meta-analysis”

– In Table 1 I would have liked to see more demographic and socioeconomic variables measured in the study. As mentioned, studies of sexual behavior are very vulnerable to confounding due to social determinants of health. While it is possible that these variables are not different in pre- and post- sex samples in this study, it is necessary to show this for the reader to assess whether there is a potential for confounding.

Thank you for the helpful comments about social determinants of health both here and throughout this review. We have added information about education, income, rural/urban residence, and roof material (an indicator of poverty) to Table 1 (page 7). These questions were only asked at the enrollment visit. In addition, we now state that all participants were Black Africans assigned female at birth (page 6). Kenyan ethnicity was not collected from participants.

– The variables included in the multivariate model are not appropriate if the objective is to adjust for confounders, as many of the variables are not confounders and are rather mediators or potentially downstream effects of immune markers. Adjusting for these would lead to a biased estimate of the causal effect of vaginal sex. The authors should take more time to reflect on the potential causal relationships between these variables and how they influence their analysis. Some reflections:– Age: is the most clear-cut confounder in this analysis and should certainly be included in the model, as age influences sexual behavior and is likely to be correlated with levels of immune markers.– Menstrual phase: is appropriate to include in the model despite not necessarily meeting the traditional definition of a confounder. The menstrual phase is unlikely to influence the initiation of sex. However, there may be chance differences in menstrual phase between samples pre- and post-sex that may warrant adjustment. As the menstrual phase is not a mediator or a downstream effect of sex, adjustment for this variable will not bias results.– Pregnancy and contraception: these variables are potential mediators on the causal pathway between initiation of sex and levels of immune markers, and should not be included in the model. If the objective of the authors is to measure only the direct (non-mediated) effect of sex, then a more complex mediation analysis would be needed to disentangle the effects of sex, pregnancy, and contraception, and the model would need to include potential confounders which are associated with pregnancy and contraception use (ex. socioeconomic variables).– Nugent score, Chlamydia infection, and HSV-2 seropositivity: these are complex variables that could be both mediators of the effects of sex but also downstream effects of changes in levels of immune markers. It is therefore problematic to adjust for these as they are potentially colliders on the causal pathway, and adjustment would potentially lead to selection (collider) bias. See the article by Hernán (Hernán A Structural Approach to Selection Bias, Epidemiology: 2004 – Volume 15 – Issue 5 – p 615-625 https://doi.org/10.1097/01.ede.0000135174.63482.43), especially figures 3-4.– Unmeasured confounders: there is potentially some residual unmeasured confounding from variables which influence sexual initiation and which also potentially influence immune response which are not included in the model. The most likely are variables related to social determinants of health (socioeconomic status, ethnicity, etc.). If the authors have access to these variables from the original study, they should consider their inclusion in the model, especially if they are strongly correlated with sexual initiation.

Thank you for these insightful comments. In response, we made the following major changes:

1) Removed the multivariable model

2) Added a new multivariable model with a new model that adjusts for age, education, income, rural/urban residence, and roof material (proxy for household wealth). The effect sizes from this new model are very similar to the effect sizes from the univariate model.

a. Results page 13: “In addition, we sought to assess whether the differences between pre- and post-first sex would be explained by social determinants of health. There was limited diversity in our cohort in these variables, in particular in terms of race and ethnicity. We adjusted the models from our primary analysis for age, education, whether the participants reported a regular source of income, whether the participants lived in an urban or rural environment, and whether the participants lived in a home with a metal or tile roof (lacking such a roof being an indicator of poverty). As shown in Figure 5AB, the effect sizes from the multivariate models were similar to those from the univariate models, suggesting that these variables do not explain the results we observed.”

b. Discussion pages 18-19: Quoted two comments above.

3) Conducted a new univariate analysis on a subset of samples that were negative for pregnancy, contraception, BV, HSV-2, and CT. Our goal with this analysis was not to precisely estimate the direct effect of beginning sexual activity in the absence of these factors, but rather to determine whether these factors solely explained our results. Essentially we felt that if the differences between pre- and post-first sex completely disappeared when we removed these samples, then our results weren’t very interesting. That finding would indicate that it was just one of those factors causing the difference between pre- and post. Because the differences don’t disappear, we conclude that those factors are not the sole cause of the differences we observe – the differences must be caused by either beginning sexual intercourse or some other unmeasured variable(s).

a. Results page 12: “We next sought to determine whether the differences between pre- and post-first sex could be explained by co-variates that differed between the groups or that are known to affect immune mediator concentrations (Table 1). In particular, pregnancy, CT infection, HSV-2 seropositivity, use of contraception, and elevated Nugent score are all known to affect cervicovaginal immune mediator concentrations. Because of this, we wished to know whether one or more of these factors was responsible for the differences we observed between pre- and post-first sex samples. To address this possibility, we repeated our analysis on a restricted sample set, excluding all samples where at the visit the participant was pregnant, positive for CT or HSV-2, reported using contraception, or had a Nugent score of 4 or greater. The analysis of these 137 samples (102 pre- and 35 post-first sex) are shown in Figure 5AB. The effects remained positive for all immune mediators in this subset analysis, indicating that the differences we saw between pre- and post-first sex samples were not solely a result of acquisition of STIs or pregnancy.”

b. Discussion page 21: “In a secondary analysis, we removed all samples where the participant used contraception, was pregnant, had a Nugent score above 3, or was positive for HSV-2 or CT. The results of this analysis was similar to our primary, univariate analysis of the full cohort. These factors have complex relationships with sexual activity: Each can be caused by sexual activity and several of these variables could also be influenced (in part) by changes in immune mediator level. Thus, they may mediate the effect of sex on immune mediator levels or they may be a consequence of it. Our goal with this secondary analysis was not to precisely estimate the direct effect of beginning sexual activity in the absence of these factors, but rather to determine whether these factors solely explained our results. Because the effects remained after removing these samples, we conclude that these factors by themselves do not solely explain the results we observed.”

4) Removed claims that we have eliminated all important confounding factors and wrote a more nuanced paragraph about causality in the Discussion (quoted two comments above).

– The strongest evidence that there is likely to be potential selection bias and unmeasured confounding in the main (unpaired) analysis comes from the paired analysis, where the point estimates for the effect sizes are much smaller than in the main analysis. I think this analysis is the strongest analysis in the paper and am surprised in fact that this was not considered as the main analysis. The authors should discuss the results from this analysis more, as the paired analysis controls for many of the unmeasured confounders that would bias the results in the main analysis, so the results from this analysis are more likely to be valid estimates of causal effect than the results from the unpaired analysis.

We have added text near the beginning of the Discussion addressing the fully-paired analysis in much more detail and in a much more balanced fashion. We describe how this analysis suggests that the analysis of the full cohort may be over-estimating the effect. In addition, we describe how the shorter time since first sex in the fully-paired subset may also contribute to the smaller effect sizes. We conclude by recommending a larger paired study in the future with longer follow-up. The paragraph reads as follows (Discussion pages 18-19):

“The fully paired subset of our cohort (where participants provided pre and post samples) had smaller effect sizes than the full cohort. Most effects remained positive, but they were smaller in magnitude than in the primary analysis and all were non-significant. A possible explanation for the smaller effects is that immune mediator concentrations increase over time post-first sex (Figure 4). The post-first sex samples in the fully paired subset were collected sooner after first sex (median 32 days) than those from the participants who provided only post-first sex samples (median 157 days). Thus, the effects may have been weaker in the fully paired subset, in part, because the samples were obtained such a short time after first sex. Another possible explanation for the smaller effects is unmeasured confounding in the full cohort. Because participants in this subset provided both pre- and post-first sex samples, the subset largely controls for unmeasured confounding factors. These unmeasured factors could have inflated the effect sizes in the full cohort. A future study with a larger number of participants followed from pre- to post-first sex and with additional specimens collected for a longer period of time post-sex would be necessary to resolve this uncertainty.”

– While the analysis for effect of time in Figure 4 is interesting, I would have liked to see p-values for the spline effect in the model. It is easy to fit different curves pre- and post-sex, but it does not necessarily follow that the addition of a spline leads to a significant improvement in the model performance. If the p-values for the spline are not significant, it would suggest the data does not support that there is an effect of time since initiating sex. Furthermore, this analysis should be described in the methods, I did not see this analysis described anywhere.

Thank you for your comments about the spline analysis in Figure 4. In the first submission, we fit separate models pre- and post-first sex. In response to this comment, we changed to single models using linear splines, with a knot at day 0 (the day of first sex). This new analysis is shown in Figure 4. In addition, we added a new table (new Table 4; page 12), including the p-values for the time-dependent spline effects (of which 17/19 are significant at p<0.05).

Results page 10-11: “As shown in Figure 4 and Table 4, immune mediator concentrations were generally stable for three years prior to first sex and then increased sharply in the year following first sex. The change in slope following first sex was statistically significant at p<0.05 for 17/19 immune mediators. This pattern is consistent with cumulative increases in immune mediator concentrations following first sex.”

In addition, we have more thoroughly described this analysis in the Methods section. Thank you for pointing out that we had neglected to describe this analysis in the Methods.

Methods page 28: “We additionally wished to assess whether immune mediator concentrations changed over time with respect to the date of first sex. To assess this question, we fit mixed-effects models with linear splines, using days since first sex as a fixed effect and participant as a random effect, with the outcome being the concentration of immune mediator (log2 pg/mL). For the splines, we used a knot at day 0, which was the day of first sex.”

– The analysis of the correlation between estimates from different models (Figure 5D) has no analytical utility and should be removed. It is expected the results from different models would be correlated as they are based on the same data. High correlation between estimates from different analyses does not mean that the results or the model structure is valid; the validity of an analysis is based on subject matter knowledge and consideration of whether the analysis is appropriate for the question at hand. This analysis is therefore not informative.

We have removed the correlations and appreciate the correction.

– The logistic and linear regressions provide different information about the data; the logistic regression informs us on whether a particular marker is more likely to be detected in post-sex samples, whereas the linear regression informs us on whether the level of a marker is higher in post-sex samples. I would have therefore been interested in seeing the results from both regression models for all of the markers, instead of showing the linear regression results for some and the logistic regression results for others.

The reason we used logistic regressions for low-detected immune mediators is that the distributions of the concentrations deviates strongly from the normal distribution when most samples are below the limit of detection. We didn’t feel that linear regression was appropriate in those cases and while logistic regression certainly provides different information, we felt that it was the best alternative. Similarly, we could not have performed logistic regression on every immune mediator, as several immune mediators were detected in every sample.

We have added a sentence to the methods explaining our rationale for this choice (page 28: “Logistic models were used for immune mediators with low levels of detection because the distributions become increasingly non-normal as more samples fall below the lower limit of detection”).

For completeness, the Author response table 1 shows the results of linear and logistic models fit on every immune mediator. The Coef columns are the log2 difference for the linear models and the log-odds for the logistic model. The direction and significance of the changes overall are similar by both models. The coefficients have the same sign whether linear or logistic for every immune mediator except CCL5. The biggest p-value differences between models appear for immune mediators which were detected in most samples, which often had low p-values by linear analysis but had higher p-values by logistic analyses (IL6, CCL3, IL1b, CXCL10, CXCL9), presumably because such a small number of samples were undetectable for those immune mediators.

**Author response table 1. sa2table1:** 

	**Linear**	**Logistic**					
Assay	Coef	Std. Error	P-value	Coef	Std. Error	P-value	Percent detectable
IL-12p70	0.24	0.14	0.08	2.02	1.03	0.05	20.0
IL-10	0.32	0.16	0.05	1.15	0.46	0.01	22.8
IFNγ	0.49	0.20	0.02	1.04	0.46	0.02	26.7
IL-17A	0.64	0.25	0.01	0.80	0.34	0.02	50.6
IL-7	0.02	0.18	0.92	0.37	0.34	0.27	50.6
TNF	0.69	0.29	0.02	0.90	0.44	0.04	68.3
IL-2	0.90	0.25	0.0005	1.28	0.52	0.01	77.8
CCL4	0.42	0.27	0.12	0.74	0.56	0.19	86.1
IL-6	0.81	0.39	0.04	1.94	2.24	0.39	89.4
CCL3	0.62	0.22	0.006	0.24	0.57	0.68	91.7
CCL5	0.39	0.28	0.17	**-7.40**	4.97	0.14	93.9
CCL20	0.53	0.36	0.15	0.65	0.83	0.43	95.6
IL-1b	1.33	0.43	0.002	0.46	0.85	0.59	96.1
CXCL10	0.74	0.32	0.02	32.92^a^	8078958	1	99.4
CXCL9	0.60	0.26	0.02	176.10^a^	8078958	1	99.4
CXCL8	1.02	0.33	0.003	^b^			100
IL-1a	0.67	0.24	0.006	^b^			100
IL-1RA	0.48	0.22	0.03	^b^			100
IL-18	0.44	0.36	0.23	^b^			100

^a^ One single pre-sex sample was undetectable and no post-sex samples were, explaining highly imprecise values.

^b^ No samples were undetectable, so logistic models could not be fit.

– I think Figure 3 is a good way to present results (multiplicative change in marker levels). I would suggest also adding the point estimates of the ratios and their confidence intervals on the right-hand side of each estimate (forest plot style), as the ratio estimates are more intuitive than the regression coefficients in Table 2. Also, the figures should distinguish between the scale of the axes (log 2) vs the units of the axes. Usually, the parentheses in the labels are used to indicate the axis units. The (log2) in the label may therefore confuse readers, as the axes do not show the effect of an increase in a log2 unit, but the ratios pre- vs. post-sex on a log2 scale. This could be better explained in the legend.

Thank you for these helpful suggestions. We made the following changes:

– Relabeled the axes and improved the legends of Figures 3, 5, and 6

– Added ratios and CIs to the side of each estimate in Figure 3

– We did not add these values to Figures 5 or 6 because they would have made the figures too complicated. The values are available in Additional File 2 for both figures as well as in the forest plots in Additional File 6 for Figure 6.

– We also added the ratios and CIs to Tables 2, 3, and 5. This way, when looking at the tables, the reader can interpret either log2 differences or ratios, whichever they find more intuitive.

– Figure 5C and 5D appear to be mislabeled.

Thank you for pointing out the error. We have corrected this mistake.

Reviewer #2 (Recommendations for the authors):– How easy technically to get CVL from AGYW pre-first sex? How is it possible to discern the impact of the sampling on the mediators?

We used an innovative sampling technique that collected CVL without use of a speculum, which was thought to be unpleasant and unnecessary in AGYW pre-first sex. We placed a small amount of IV tubing in the fornix and instilled saline in that way, this was well tolerated. We have added a sentence to the Strengths and limitations section of the Discussion noting that this method made it technically possible to perform the study (page 20: “Our sampling technique allowed collection of CVL without a speculum exam, which improved the feasibility of this research in a population with limited sexual experience”).

In addition, we sought to recruit from a broad population. Our study used a community recruitment strategy and did not recruit from adolescents seeking clinical care, therefore we might have recruited youth who were uncomfortable with clinical procedures. We used the meta-analysis as a further way to verify results from outside of our own cohort to try to assess whether these results were applicable in other geographic populations and with different laboratories. We feel satisfied that our cohort results are similar to other cohorts in our meta-analysis.

– Why exclude STIs if these can be controlled for in the MV models?

We excluded 5 samples due to rare STIs because we felt we could not accurately control for them in the multivariate models given the small sample sizes (n=1-2 per STI). We clarified this point in the Results section. However, we included samples with more common (in our cohort) STIs.

Excluded: Trichomonas (1 sample), Gonorrhea (1 sample), active genital HSV-1 infection (1 sample), active genital HSV-2 infection (2 samples), where “active” means DNA was detected in a genital sample.

Included: Chlamydia, HSV-1 seropositive, HSV-2 seropositive, (samples with BV were also included)

Also unclear why differentiating sexual activity and STIs is so critical if the effects of both often co-occur after sexual activity is initiated. Perhaps this makes sense since STIs were rare, but it seems like it could affect generalizability.

Our goal was to determine if sexual activity itself affects the genital immune milieu. For that reason, we felt it was important to determine whether any changes were solely a result of STIs. If we had observed changes in the univariate model that disappeared after controlling for STIs, we would have concluded that cytokine changes post-first sex were only a result of STIs. Instead, we observed that the changes persisted, suggesting that they are associated with sexual activity itself (or some other unknown factor), but not specifically the STIs present in the cohort.

– The sensitivity analysis of participants sampled only pre- and post- sex is a strength, as this comparison will likely reduce heterogeneity between individuals. Here the focus should be less on P-values and more on consistency between effect sizes. A correlation is presented, but this could be heterogeneous because of estimates being stronger or weaker in the paired analysis. Therefore it may make more sense to compare these on a case-by-case basis, depending on the consistency in these trends.

Thank you for these comments. As also discussed in the response to Reviewer 1, we have removed the correlations. We have substantially expanded our discussion of this analysis and focused on comparing effect sizes in our interpretation. We have additionally tempered our claims of causality based on this analysis.

– Abstract- doesn't make sense to say a "median" 54% increase, as the % increase will mean something different for different markers, depending on their individual distributions. Markers with wider distributions will naturally have great % increases.

We have removed this passage from the abstract.

– Re the 57 samples reported after first sex, and the 12 samples deemed as such due to pregnancy, presence of semen biomarkers, STIs etc- is the assumption that the 12 samples did not overlap with the 57, e.g. represent under-reporting of first sex?

Thank you for pointing out that this issue was confusing. We added a sentence to the Results clarifying that there were a total of 69 post-first sex samples (57 reported and 12 inferred based on test results). The relevant section (page 5) now reads:

“Of these, 111 samples from 75 participants were classified as pre-first sex and 57 samples from 45 participants were obtained after reported first sex. In addition, 12 samples from 8 participants were classified as post-sex due to presence of Y-chromosome DNA, PSA or because of pregnancy or a positive *Chlamydia trachomatis* (CT) test at that visit or a prior visit. Thus, there were a total of 69 post-sex specimens from 53 participants.”

The (important!) issue of underreporting is discussed in the Strengths and limitations section of the Discussion:

Page 20-21: “An important limitation of our study is possible misclassification of specimens from pre-sexual activity. It is important to note that all three cohorts presented here were specifically designed to assess the sexual activity of adolescents and used best practices in ascertaining this challenging key variable. Of the 123 samples where participants reported no sexual activity, we classified 12 (9.8%) as post-first sex based on STI, pregnancy, Y-chromosome, or PSA results. Misclassification of specimens as pre/post-first sex is a challenge in this field, which we did our best to mitigate, but residual misclassification is possible.”.

– In figure 5, even though most covariates didn't affect the estimates, would some of these be considered possible "mediators" as opposed to "confounders". E.g. BV has been associated with lower CXCL10 levels, so changes in BV post sex (which has also been shown) could lead to decreases in some of these key chemokines. This is worth considering, in particular since participants were sampled for the most part a long time before/after sex. It does appear in the Discussion. Is there clear enough evidence to conclude that the effects are unlikely to be due to sex alone? The Discussion of this on page 16 re: CXCL10 does not seem to match how the data are presented.

Thank you for your comments about certain co-variates being mediators rather than confounders. This issue was also raised by Reviewer 1. We have substantially changed our multivariate model and reworked the presentation and discussion of results pertaining to the covariates. We additionally added a new analysis of a subset of the full cohort containing only the subset of samples that were negative for pregnancy, contraception, BV, HSV-2, and CT. Because the effects remain positive in that subset, we conclude that these factors are not solely causing the effects we see. Rather, the effects likely derive from sex itself as well as possible unmeasured variables.

We removed the Discussion of CXCL10 because it derived from the earlier multivariable model, which is no longer in the paper. We agree with you about CXCL10: BV decreases CXCL10 and BV increases post-first sex. Though we still see an increase in CXCL10 post-first sex, BV is probably reducing its magnitude compared to the increase that would be seen without BV. Thoroughly disentangling these interactions is beyond the scope of this paper and set of samples.

– Adjusting for protein concentration- certainly worth reporting this increases post-sex, but could this be due to general increases in inflammation following sex.

Thank you for this suggestion. We have added it to the Discussion on page 21: “It is possible that the increases in total protein concentration post-first sex are associated with the general increases in immune mediators observed post-first sex.”

– Bottom of page 12, paragraph starting with "The specimens from the fully-paired…" – it is unclear what the point of this paragraph is.

We agree that the purpose of this paragraph was not clear as written. The purpose was to discuss the comparability of specimens from the fully-paired subset to the rest of the specimens and we have rephrased the paragraph to make that clear (now on page 15).

“The specimens from the fully-paired subset were generally comparable to the specimens from the participants who only provided post specimens, with one important difference: the post specimens in the fully-paired subset were obtained sooner after reported date of first sex (p = 4.7E-8). In the fully-paired subset, the post-first sex samples came only about a month after first sex (median 31.5 days). In contrast, the post-first sex samples from the other participants came about four months after first sex (median 157 days). The age of the participants at the time of first sex, however, was similar (18.9 years in each subset, p = 0.90).”